# LoRA Done RITE: Robust Invariant Transformation Equilibration for LoRA Optimization

**Jui-Nan Yen** [*1]    **Si Si** [2]    **Zhao Meng** [2]    **Felix Yu** [2]    **Sai Surya Duvvuri** [*3]

**Inderjit S. Dhillon** [2]    **Cho-Jui Hsieh** [12]    **Sanjiv Kumar** [2]

## Abstract

Low-rank adaption (LoRA) is a widely used parameter-efficient finetuning method for LLMs that reduces memory requirements. However, current LoRA optimizers lack transformation invariance, which leads to weight updates that depend on how the two LoRA factors are scaled or rotated. This deficiency leads to inefficient learning and sub-optimal solutions in practice. This paper introduces LoRA-RITE, a novel adaptive matrix preconditioning method for LoRA optimization, which achieves transformation invariance while being computationally efficient. We provide theoretical analysis to demonstrate the benefit of our method and conduct experiments on various LLM tasks with different models including Gemma 2B, 7B, and mT5-XXL. The results demonstrate consistent improvements over existing optimizers. For example, replacing Adam with LoRA-RITE during LoRA fine-tuning of Gemma-2B yields 4.6% accuracy gain on Super-Natural Instructions and 3.5% accuracy gain across four other LLM benchmarks (HellaSwag, ArcChallenge, GSM8K, OpenBookQA).

## 1 Introduction

Low-Rank Adaptation (LoRA) (Hu et al., 2022) is a popular parameter-efficient method for fine-tuning Large Language Models (LLMs). By freezing the pretrained weights and injecting trainable low-rank matrices into each layer, LoRA significantly reduces memory requirements and mitigates overfitting in limited data settings. More formally, letting $W \in \mathbb{R}^{m \times n}$ be a weight matrix in an LLM, LoRA freezes $W$ and introduces a low-rank matrix $Z$ added to $W$, where $Z$ is represented by the multiplication of two rank-$r$ matrices $A$ and $B$, i.e.,

$$Z = AB^\top \in \mathbb{R}^{m \times n}, A \in \mathbb{R}^{m \times r}, B \in \mathbb{R}^{n \times r}, \ r \ll \min(m, n). \tag{1}$$

The matrices $A$ and $B$ will be referred to as LoRA factors in this paper. Recent research has explored numerous variations and improvements over the classic LoRA algorithm (Valipour et al., 2023; Zhang et al., 2023b; Liu et al., 2024; Yaras et al., 2024).

Despite being widely used in practice, we find that applying standard optimizers to LoRA leads to updates that are not "transformation invariant". By definition of LoRA in (1), the same update $Z$ can be decomposed in multiple ways, i.e., $Z = A_1 B_1^\top = A_2 B_2^\top$. Ideally, an optimizer should yield the same update to $Z$ regardless of the specific factorization. However, commonly used optimizers with diagonal preconditioners like Adam (Kingma & Ba, 2014), Adagrad (Duchi et al., 2011), RMSProp (Tieleman & Hinton, 2012), and even second-order methods like Shampoo (Gupta et al., 2018) and CASPR (Duvvuri et al., 2024), violate this principle when applied to LoRA. This violation not only presents a mathematical inconsistency but also leads to significant inefficiencies during training. In practice, we observe that one LoRA factor often dominates the optimization process, receiving substantial updates while the other remains nearly fixed. Although this can be partially mitigated by some recently proposed approaches such as employing different learning rates for the two factors (Hayou et al., 2024), we ask the question: *is there a more principled way to design an optimizer that inherently enforces transformation invariance for LoRA?*

To address this challenge, we first prove that any form of diagonal preconditioner cannot achieve transformation invariance, which motivates the use of matrix preconditioners. However, existing

---

*Work done while at Google. [1]UCLA [2]Google [3]UT Austin

matrix preconditioners like Shampoo and CASPR lack transformation invariance and introduce significant computational and memory overhead. To overcome these limitations, we propose LoRA-RITE (Robust Invariant Transformation Equilibration), a novel optimizer designed specifically for LoRA optimization. LoRA-RITE employs a transformation-invariant preconditioner on the low-rank side, achieving transformation invariance without incurring substantial overhead. Furthermore, we demonstrate how to maintain this property when incorporating first and second moments, crucial for the practical effectiveness of adaptive optimization methods. Empirical evaluations across various datasets and models confirm the effectiveness of the proposed algorithm.

The contribution of this paper can be summarized below:

- We propose LoRA-RITE, the first adaptive matrix preconditioning optimizer for LoRA that is transformation-invariant, the property that is lacking in most existing optimizers when applied to LoRA. Theoretically, we provide a convergence analysis for our method.
- Despite utilizing matrix preconditioners, LoRA-RITE achieves little overhead in both memory and time compared to first-order optimizers, especially when the LoRA rank ($r$) is significantly smaller than the original matrix dimensions $m$ and $n$.
- The proposed optimizer leads to significantly improved performance across multiple datasets and architectures. For instance, when applied to the GSM8K (Cobbe et al., 2021) dataset with a Gemma 7B IT model (Gemma Team et al., 2024), LoRA-RITE achieves a 55.50% accuracy rate. This surpasses the widely-used Adam optimizer (Kingma & Ba, 2014) by a substantial margin (48.37%) and even outperforms the second-best optimizer on this dataset, Lamb (You et al., 2020) (50.64%), by approximately 5%.

## 2 TRANSFORMATION INVARIANCE FOR LORA OPTIMIZATION

We now introduce the concept of transformation invariance in LoRA training and demonstrate that most existing optimizers, when applied to LoRA, do not satisfy this property. This deficiency leads to inefficient learning in practice.

### 2.1 DEFINITION OF TRANSFORMATION INVARIANCE

As introduced in (1), LoRA adds a low-rank matrix $Z = AB^\top$ to the original weight matrix $W$ and learns the LoRA factors $A \in \mathbb{R}^{m \times r}, B \in \mathbb{R}^{n \times r}$ to minimize the fine-tuning loss. Observe that many different LoRA factors $(A_1, B_1), (A_2, B_2)$ can represent the same finetuned weight,

$$Z = A_1 B_1^\top = A_2 B_2^\top. \tag{2}$$

When an optimizer is applied to train LoRA, it will produce different updates, $\delta A_1, \delta B_1$ or $\delta A_2, \delta B_2$, based on the specific parameterization used. Even though $(A_1, B_1)$ and $(A_2, B_2)$ represent the same finetuned weight $Z$, the updates using different parameterizations can produce different updates to $Z$. This suggests a serious inconsistency and implies that the update could be suboptimal under some parameterizations. Based on this observation, we propose that LoRA optimization should ensure *transformation invariance*, defined as follows:

**Definition 1** (Transformation Invariance). *Let $(A_1, B_1)$ be a pair of LoRA factors and let $A_2 = A_1 R, B_2 = B_1 R^{-\top}$ for some invertible matrix $R$. An optimizer exhibits transformation invariance if its updates, $(\delta A_1, \delta B_1)$ and $(\delta A_2, \delta B_2)$, satisfy*

$$(A_1 + \delta A_1)(B_1 + \delta B_1)^\top = (A_2 + \delta A_2)(B_2 + \delta B_2)^\top := Z + \delta Z. \tag{3}$$

This means the optimizer should produce the same update, $\delta Z$, to the fine-tuned weights for any equivalent LoRA factorizations. As a special case, we introduce scalar scale invariance below in Definition 2 as a weaker version of transformation invariance, which only requires that updates remain equivalent when the LoRA factors are scaled up or down by a scalar factor. Formally, we define it as:

**Definition 2** (Scalar Scale Invariance). *Let $(A_1, B_1)$ be a pair of LoRA factors and let $A_2 = sA_1, B_2 = (1/s)B_1$ for some nonzero scalar constant $s$. An optimizer exhibits scalar scale invariance if its updates, $(\delta A_1, \delta B_1)$ and $(\delta A_2, \delta B_2)$, satisfy*

$$(A_1 + \delta A_1)(B_1 + \delta B_1)^\top = (A_2 + \delta A_2)(B_2 + \delta B_2)^\top.$$

Surprisingly, we will show that most commonly used LoRA optimizers do not even satisfy this weaker form of transformation invariance.

## 2.2 EXISTING OPTIMIZERS ARE NOT SCALAR SCALE INVARIANT

We now show that neither gradient descent nor Adam are scalar scale invariant, and in fact, almost all the existing optimizers are not scalar scale invariant when applied to LoRA optimization.

Assume loss is $f(\boldsymbol{Z})$, and the gradient of the loss function $f$ with respect to $\boldsymbol{Z}$, $\boldsymbol{A}_1$, $\boldsymbol{B}_1$, $\boldsymbol{A}_2$, $\boldsymbol{B}_2$ to be $\nabla \boldsymbol{Z} := \partial f/\partial \boldsymbol{Z}$, $\nabla \boldsymbol{A}_1 := \partial f/\partial \boldsymbol{A}_1$, $\nabla \boldsymbol{B}_1 := \partial f/\partial \boldsymbol{B}_1$, $\nabla \boldsymbol{A}_2 := \partial f/\partial \boldsymbol{A}_2$, and $\nabla \boldsymbol{B}_2 := \partial f/\partial \boldsymbol{B}_2$ respectively. Recall $\boldsymbol{Z} = \boldsymbol{A}_1\boldsymbol{B}_1^\top = \boldsymbol{A}_2\boldsymbol{B}_2^\top$ and by chain rule, we have

$$\nabla \boldsymbol{A}_1 := \partial f/\partial \boldsymbol{A}_1 = \partial f/\partial \boldsymbol{Z} * \partial \boldsymbol{Z}/\partial \boldsymbol{A}_1 = \partial f/\partial \boldsymbol{Z} * \boldsymbol{B}_1 = \nabla \boldsymbol{Z}\boldsymbol{B}_1, \qquad (4)$$

Similarly, we have $\nabla \boldsymbol{B}_1 = \nabla \boldsymbol{Z}^\top \boldsymbol{A}_1$, $\nabla \boldsymbol{A}_2 = \nabla \boldsymbol{Z}\boldsymbol{B}_2$, and $\nabla \boldsymbol{B}_2 = \nabla \boldsymbol{Z}^\top \boldsymbol{A}_2$. For gradient descent,

$$\delta \boldsymbol{A}_1 := -\eta\nabla \boldsymbol{A}_1, \ \delta \boldsymbol{B}_1 := -\eta\nabla \boldsymbol{B}_1, \ \delta \boldsymbol{A}_2 := -\eta\nabla \boldsymbol{A}_2, \ \delta \boldsymbol{B}_2 := -\eta\nabla \boldsymbol{B}_2,$$

where $\eta$ is the learning rate. To test scalar scale invariant, let $\boldsymbol{A}_2 = s\boldsymbol{A}_1$, $\boldsymbol{B}_2 = (1/s)\boldsymbol{B}_1$, we have

$$\delta \boldsymbol{A}_2 = -\eta\nabla \boldsymbol{Z}\boldsymbol{B}_2 = -(1/s)\eta\nabla \boldsymbol{Z}\boldsymbol{B}_1 = (1/s)\delta \boldsymbol{A}_1 \quad (\text{since } \boldsymbol{B}_2 = (1/s)\boldsymbol{B}_1) \qquad (5)$$

$$\delta \boldsymbol{B}_2 = -\eta\nabla \boldsymbol{Z}^\top \boldsymbol{A}_2 = -s\eta\nabla \boldsymbol{Z}^\top \boldsymbol{A}_1 = s\delta \boldsymbol{B}_1 \quad (\text{since } \boldsymbol{A}_2 = s\boldsymbol{A}_1). \qquad (6)$$

Consequently, from (5) and (6):

$$\boldsymbol{A}_2\boldsymbol{B}_2^\top + \delta \boldsymbol{A}_2\boldsymbol{B}_2^\top + \boldsymbol{A}_2\delta \boldsymbol{B}_2^\top + \delta \boldsymbol{A}_2\delta \boldsymbol{B}_2^\top = \boldsymbol{A}_1\boldsymbol{B}_1^\top + (1/s^2)\delta \boldsymbol{A}_1\boldsymbol{B}_1^\top + s^2\boldsymbol{A}_1\delta \boldsymbol{B}_1^\top + \delta \boldsymbol{A}_1\delta \boldsymbol{B}_1^\top$$
$$\neq \boldsymbol{A}_1\boldsymbol{B}_1^\top + \delta \boldsymbol{A}_1\boldsymbol{B}_1^\top + \boldsymbol{A}_1\delta \boldsymbol{B}_1^\top + \delta \boldsymbol{A}_1\delta \boldsymbol{B}_1^\top,$$

and so (3) does not hold for arbitrary $s$. Therefore, gradient descent is not scalar scale invariant, and the gradient can be arbitrary large for the LoRA factors when $s$ goes to $0$ or infinity.

Can this issue be mitigated by adaptive updates such as Adam? The answer is no. To see this, let

$$\delta \boldsymbol{A}_1 := -\eta\nabla \boldsymbol{A}_1/(\nabla \boldsymbol{A}_1 \odot \nabla \boldsymbol{A}_1)^{1/2}$$

be the Adam update for $\boldsymbol{A}_1$, where $\odot$ denotes elementwise multiplication. Note that we omit the momentum term for brevity. Defining updates of $\boldsymbol{A}_2$, $\boldsymbol{B}_1$ and $\boldsymbol{B}_2$ similarly, we have

$$\delta \boldsymbol{A}_2 = -\frac{\eta\nabla \boldsymbol{A}_2}{(\nabla \boldsymbol{A}_2 \odot \nabla \boldsymbol{A}_2)^{1/2}} = -\frac{(1/s)\eta\nabla \boldsymbol{A}_1}{(1/s)(\nabla \boldsymbol{A}_1 \odot \nabla \boldsymbol{A}_1)^{1/2}} = \delta \boldsymbol{A}_1$$

and similarly $\delta \boldsymbol{B}_2 = \delta \boldsymbol{B}_1$. As a result,

$$\boldsymbol{A}_2\boldsymbol{B}_2^\top + \delta \boldsymbol{A}_2\boldsymbol{B}_2^\top + \boldsymbol{A}_2\delta \boldsymbol{B}_2^\top + \delta \boldsymbol{A}_2\delta \boldsymbol{B}_2^\top = \boldsymbol{A}_1\boldsymbol{B}_1^\top + (1/s)\delta \boldsymbol{A}_1\boldsymbol{B}_1^\top + s\boldsymbol{A}_1\delta \boldsymbol{B}_1^\top + \delta \boldsymbol{A}_1\delta \boldsymbol{B}_1^\top$$
$$\neq \boldsymbol{A}_1\boldsymbol{B}_1^\top + \delta \boldsymbol{A}_1\boldsymbol{B}_1^\top + \boldsymbol{A}_1\delta \boldsymbol{B}_1^\top + \delta \boldsymbol{A}_1\delta \boldsymbol{B}_1^\top,$$

thus failing to satisfy scalar scale invariance. Actually, one can see that most of the existing optimizers, such as Adagrad (Duchi et al., 2011), RMSProp (Tieleman & Hinton, 2012), and Shampoo (Gupta et al., 2018) are not scalar scale or transformation invariant.

## 2.3 BENEFITS OF TRANSFORMATION INVARIANCE

Why is transformation invariance important? Beyond the mathematical argument that different parameterizations of the same weight update should be equivalent, we demonstrate that transformation invariance leads to more efficient feature learning. The concept of efficient feature learning, introduced in (Hayou et al., 2024), describes the asymptotic training behavior of LoRA as the network width grows. As discussed earlier, for LoRA, the update to the matrix $\boldsymbol{Z} = \boldsymbol{A}\boldsymbol{B}^\top$ can be decomposed into three parts

$$\delta \boldsymbol{Z} = (\boldsymbol{A} + \delta \boldsymbol{A})(\boldsymbol{B}^\top + \delta \boldsymbol{B}^\top) - \boldsymbol{A}\boldsymbol{B}^\top = \delta \boldsymbol{A}\boldsymbol{B}^\top + \boldsymbol{A}\delta \boldsymbol{B}^\top + \delta \boldsymbol{A}\delta \boldsymbol{B}^\top,$$

where the $\delta \boldsymbol{A}\delta \boldsymbol{B}^\top$ is typically negligible as it depends on the square of the learning rate. Efficient feature learning requires that both $\delta \boldsymbol{A}\boldsymbol{B}^\top \boldsymbol{x}$ and $\boldsymbol{A}\delta \boldsymbol{B}^\top \boldsymbol{x}$ are of magnitude $\mathcal{O}(n^0) = \mathcal{O}(1)$ with respect to the network width $n$, where $\boldsymbol{x}$ is the input embedding. On the contrary, if the magnitude of $\delta \boldsymbol{A}\boldsymbol{B}^\top \boldsymbol{x}$ is dependent on $n$: $\mathcal{O}(n^\alpha)$, then it can explode if $\alpha > 0$ and diminish if $\alpha < 0$, as the network width $n$ grows.

Hayou et al. (2024) show that conventional optimizers do not satisfy efficient feature learning. This can be seen from Figure 1, where the weight norm for factor $\boldsymbol{B}$ changes significantly while the weight norm for factor $\boldsymbol{A}$ barely changes. More discussions about the this unbalanced training dynamic are in Appendix A.8.

Under mild assumptions, we can show that a transformation-invariant optimizer guarantees efficient feature learning. The proof is given in Appendix A.3.

**Theorem 1.** *Any optimizer that is transformation-invariant and uses the same update rule for both* $\boldsymbol{A}$ *and* $\boldsymbol{B}$ *will achieve efficient feature learning.*

Beyond the efficient learning guarantee, in practice when training LoRA with existing optimizers, it is often the case that only one of the LoRA factors is updated properly, while the other remains almost unchanged, as shown in Figure 1. This is also a consequence of lacking scalar scale invariance, as the initial scales for the two LoRA factors can be very different (one it typically initialized from 0 while other from Gaussian random).

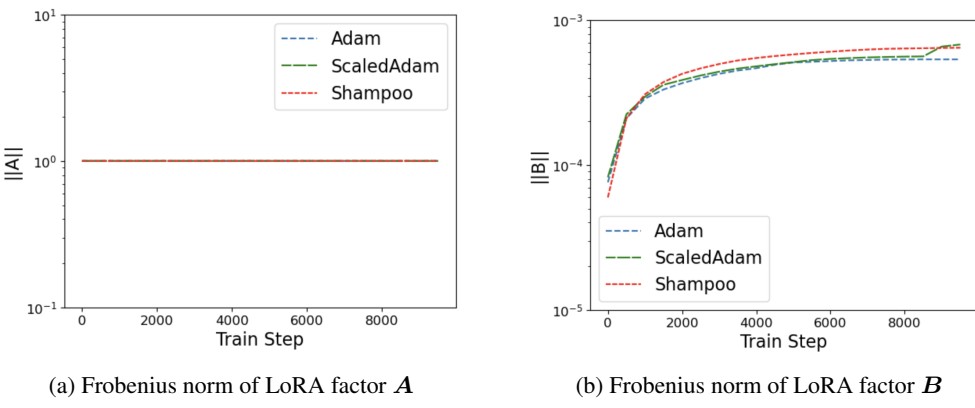

(a) Frobenius norm of LoRA factor $\boldsymbol{A}$          (b) Frobenius norm of LoRA factor $\boldsymbol{B}$

Figure 1: The norms of the LoRA factors $\boldsymbol{A}$ and $\boldsymbol{B}$ change differently across different training steps.

## 3   OUR PROPOSED OPTIMIZER

We now present our proposed algorithm, which satisfies transformation invariance, and achieves significant improvements over previous LoRA optimizers in many empirical tasks.

### 3.1   DIAGONAL PRECONDITIONING IS NOT ENOUGH FOR TRANSFORMATION INVARIANCE

In this section we show that diagonal preconditioning is not enough to achieve transformation invariance. Recall that the LoRA factors are $\boldsymbol{A} \in \mathbb{R}^{m \times r}$ and $\boldsymbol{B} \in \mathbb{R}^{n \times r}$. When updating $\boldsymbol{A}$, most existing optimizers utilize the following update:

$$\text{vec}(\delta \boldsymbol{A}) = -\eta \boldsymbol{X} \, \text{vec}(\nabla \boldsymbol{A}), \tag{7}$$

where $\text{vec}(\cdot)$ lists the elements of a matrix as a vector in column-major order and $\boldsymbol{X} \in \mathbb{R}^{mr \times mr}$ is a symmetric positive definite preconditioning matrix. Diagonal preconditioning methods like Adam and Adagrad assume $\boldsymbol{X}$ is a diagonal matrix, while matrix preconditioning methods such as Shampoo and CASPR use non-diagonal $\boldsymbol{X}$.

Consider the LoRA pairs $(\boldsymbol{A}_1, \boldsymbol{B}_1)$ and $(\boldsymbol{A}_2, \boldsymbol{B}_2)$ such that: $\boldsymbol{A}_2 = \boldsymbol{A}_1 \boldsymbol{R}$, $\boldsymbol{B}_2 = \boldsymbol{B}_1 \boldsymbol{R}^{-\top}$, where $\boldsymbol{R}$ is an invertible matrix. By (4), the gradient with respect to the LoRA factors are:

$$\nabla \boldsymbol{A}_2 = \nabla \boldsymbol{Z} \boldsymbol{B}_2 = \nabla \boldsymbol{A}_1 \boldsymbol{R}^{-\top}, \quad \nabla \boldsymbol{B}_2 = \nabla \boldsymbol{Z}^\top \boldsymbol{A}_2 = \nabla \boldsymbol{B}_1 \boldsymbol{R}. \tag{8}$$

Let us consider the case when $n = m = 1$. For $\boldsymbol{A} \in \mathbb{R}^{1 \times r}$, we can substitute $\text{vec}(\delta \boldsymbol{A}) = \delta \boldsymbol{A}^\top$ and $\text{vec}(\nabla \boldsymbol{A}) = \nabla \boldsymbol{A}^\top$ in update (7) and apply transpose on both sides to get:

$$\delta \boldsymbol{A} = -\eta \nabla \boldsymbol{A} \boldsymbol{X}^\top = -\eta \nabla \boldsymbol{Z} \boldsymbol{B} \boldsymbol{X}, \quad \delta \boldsymbol{B} = -\eta \nabla \boldsymbol{B} \boldsymbol{X}^\top = -\eta \nabla \boldsymbol{Z}^\top \boldsymbol{A} \boldsymbol{X}^\top \tag{9}$$

In this case, if we have two equivalent LoRA pairs $(\boldsymbol{A}_1, \boldsymbol{B}_1), (\boldsymbol{A}_2, \boldsymbol{B}_2)$ with their corresponding preconditioners $\boldsymbol{X}_1, \boldsymbol{X}_2$, then from (8) and (9), we have

$$\delta \boldsymbol{A}_1 \boldsymbol{B}_1^\top + \boldsymbol{A}_1 \delta \boldsymbol{B}_1^\top + \delta \boldsymbol{A}_1 \delta \boldsymbol{B}_1^\top = -\eta \nabla \boldsymbol{Z} \boldsymbol{B}_1 \boldsymbol{X}_1^\top \boldsymbol{B}_1^\top - \eta \boldsymbol{A}_1 \boldsymbol{X}_1 \boldsymbol{A}_1^\top \nabla \boldsymbol{Z} + \eta^2 \nabla \boldsymbol{Z} \boldsymbol{B}_1 \boldsymbol{X}_1^\top \boldsymbol{X}_1 \boldsymbol{A}_1^\top \nabla \boldsymbol{Z} \tag{10}$$

and

$$\delta \boldsymbol{A}_2 \boldsymbol{B}_2^\top + \boldsymbol{A}_2 \delta \boldsymbol{B}_2^\top + \delta \boldsymbol{A}_2 \delta \boldsymbol{B}_2^\top = -\eta \nabla \boldsymbol{Z} \boldsymbol{B}_2 \boldsymbol{X}_2^\top \boldsymbol{B}_2^\top - \eta \boldsymbol{A}_2 \boldsymbol{X}_2 \boldsymbol{A}_2^\top \nabla \boldsymbol{Z} + \eta^2 \nabla \boldsymbol{Z} \boldsymbol{B}_2 \boldsymbol{X}_2^\top \boldsymbol{X}_2 \boldsymbol{A}_2^\top \nabla \boldsymbol{Z}$$

$$= -\eta \nabla \boldsymbol{Z} \boldsymbol{B}_1 (\boldsymbol{R}^{-\top} \boldsymbol{X}_2^\top \boldsymbol{R}^{-1}) \boldsymbol{B}_1^\top - \eta \boldsymbol{A}_1 (\boldsymbol{R} \boldsymbol{X}_2 \boldsymbol{R}^\top) \boldsymbol{A}_1^\top \nabla \boldsymbol{Z} + \eta^2 \nabla \boldsymbol{Z} \boldsymbol{B}_1 \boldsymbol{R}^{-\top} \boldsymbol{X}_2^\top \boldsymbol{X}_2 \boldsymbol{R}^\top \boldsymbol{A}_1^\top \nabla \boldsymbol{Z}. \tag{11}$$

For (10) and (11) to be equal for any arbitrary $\eta$, one can see that it is necessary to have

$$\nabla \boldsymbol{Z} \boldsymbol{B}_1 \boldsymbol{X}_1^\top \boldsymbol{X}_1 \boldsymbol{A}_1^\top \nabla \boldsymbol{Z} = \nabla \boldsymbol{Z} \boldsymbol{B}_1 \boldsymbol{R}^{-\top} \boldsymbol{X}_2^\top \boldsymbol{X}_2 \boldsymbol{R}^\top \boldsymbol{A}_1^\top \nabla \boldsymbol{Z}, \tag{12}$$

as it is the only term quadratic to $\eta$. However, there might not exist a pair of diagonal preconditioners $\boldsymbol{X}_1$, $\boldsymbol{X}_2 \neq \boldsymbol{0}$ such that the above condition is satisfied for arbitrary $\nabla \boldsymbol{Z}$, $\boldsymbol{A}$, and $\boldsymbol{B}$, because $\boldsymbol{R}$ can be a full matrix chosen in an adversarial fashion, such that $\boldsymbol{R}^{-\top} \boldsymbol{X}_2^\top \boldsymbol{X}_2 \boldsymbol{R}^\top$ is non-diagonal and thus different from the left hand side. Consequently, we conclude it is necessary to adopt matrix preconditioning to achieve transformation invariance.

## 3.2 Achieving Transformation Invariance

To achieve transformation invariance, we begin by recognizing that the LoRA factors, $\boldsymbol{A}$ and $\boldsymbol{B}$, can be decomposed into their respective orthogonal bases and magnitudes:

$$\boldsymbol{A} = \boldsymbol{U_A} \boldsymbol{R_A}, \quad \boldsymbol{B} = \boldsymbol{U_B} \boldsymbol{R_B},$$

where $\boldsymbol{U_A}$ and $\boldsymbol{U_B}$ can be obtained through polar decomposition.

Note that the gradients of $\boldsymbol{A}$ and $\boldsymbol{B}$, $\nabla \boldsymbol{A} = \nabla \boldsymbol{Z} \boldsymbol{B}$ and $\nabla \boldsymbol{B} = \nabla \boldsymbol{Z}^\top \boldsymbol{A}$, depend on both the basis and the magnitude. To achieve transformation invariance, we introduce the concept of "unmagnified gradients" distinguished from standard gradients by the symbol $\bar{\nabla}$:

$$\bar{\nabla} \boldsymbol{A} \coloneqq \nabla \boldsymbol{Z} \boldsymbol{U_B} = \nabla \boldsymbol{A} \boldsymbol{R}_{\boldsymbol{B}}^\dagger, \quad \bar{\nabla} \boldsymbol{B} \coloneqq \nabla \boldsymbol{Z}^\top \boldsymbol{U_A} = \nabla \boldsymbol{B} \boldsymbol{R}_{\boldsymbol{A}}^\dagger, \tag{13}$$

where $\boldsymbol{R}_{\boldsymbol{A}}^\dagger$ and $\boldsymbol{R}_{\boldsymbol{B}}^\dagger$ are the pseudo-inverse of $\boldsymbol{R_A}$ and $\boldsymbol{R_B}$ respectively. These unmagnified gradients, relying solely on the column spaces of $\boldsymbol{A}$ and $\boldsymbol{B}$, remain invariant to transformations of the LoRA factors. This invariance forms the cornerstone of our algorithm's ability to achieve transformation invariance.

Adaptive preconditioning methods like Adam have demonstrated superiority over non-adaptive methods like SGD. Furthermore, as established earlier, matrix preconditioning is crucial for achieving transformation invariance. Therefore, we propose utilizing these unmagnified gradients for adaptive matrix preconditioning. Additionally, we only do one-sided preconditioning, on the shorter side that is of size $r$, to ensure low time and memory complexity of the proposed method.

Since our update rule is symmetric for $\boldsymbol{A}$ and $\boldsymbol{B}$, for brevity, from now on we only describe the update rule for $\boldsymbol{A}$. For simplicity, let's first discuss the case without momentum. We propose the update rule for

$$\text{LoRA-RITE:} \quad \delta \boldsymbol{A} = -\eta \bar{\nabla} \boldsymbol{A} (\bar{\nabla} \boldsymbol{A}^\top \bar{\nabla} \boldsymbol{A})^{-1/2} (\boldsymbol{R}_{\boldsymbol{B}}^\top)^\dagger \tag{14}$$

where $\eta$ is the learning rate. This update can be broken down into two parts. The first part

$$-\eta \bar{\nabla} \boldsymbol{A} (\bar{\nabla} \boldsymbol{A}^\top \bar{\nabla} \boldsymbol{A})^{-1/2}$$

resembles the adaptive preconditioning mechanism in Adagrad, but employs matrix operations instead of element-wise operations. Crucially, the use of unmagnified gradients ensures this term remains consistent across all equivalent LoRA pairs, up to the choice of the basis.

The second part $(\boldsymbol{R}_{\boldsymbol{B}}^\top)^\dagger$ adjusts the magnitude of the update for different LoRA pairs. Since $(\boldsymbol{R}_{\boldsymbol{B}}^\top)^\dagger \boldsymbol{B}^\top = \boldsymbol{U}_{\boldsymbol{B}}^\top$, this effectively takes out the magnitude of $\boldsymbol{B}^\top$ in $\delta \boldsymbol{A} \boldsymbol{B}^\top$. We thus have

$$\begin{aligned} \delta \boldsymbol{A}_2 &= -\eta \bar{\nabla} \boldsymbol{A}_2 (\bar{\nabla} \boldsymbol{A}_2^\top \bar{\nabla} \boldsymbol{A}_2)^{-1/2} (\boldsymbol{R}_{\boldsymbol{B}_2}^\top)^\dagger = -\eta \bar{\nabla} \boldsymbol{A}_2 (\bar{\nabla} \boldsymbol{A}_2^\top \bar{\nabla} \boldsymbol{A}_2)^{-1/2} \boldsymbol{U}_{\boldsymbol{B}_2}^\top (\boldsymbol{B}_2^\top)^\dagger \\ &= -\eta \nabla \boldsymbol{Z} \boldsymbol{U}_{\boldsymbol{B}_2} (\bar{\nabla} \boldsymbol{A}_2^\top \bar{\nabla} \boldsymbol{A}_2)^{-1/2} \boldsymbol{U}_{\boldsymbol{B}_2}^\top (\boldsymbol{B}_2^\top)^\dagger = -\eta \nabla \boldsymbol{Z} (\boldsymbol{U}_{\boldsymbol{B}_2} \bar{\nabla} \boldsymbol{A}_2^\top \bar{\nabla} \boldsymbol{A}_2 \boldsymbol{U}_{\boldsymbol{B}_2}^\top)^{-1/2} (\boldsymbol{B}_2^\top)^\dagger \\ &= -\eta \nabla \boldsymbol{Z} (\boldsymbol{U}_{\boldsymbol{B}_1} \bar{\nabla} \boldsymbol{A}_1^\top \bar{\nabla} \boldsymbol{A}_1 \boldsymbol{U}_{\boldsymbol{B}_1}^\top)^{-1/2} (\boldsymbol{B}_2^\top)^\dagger = \delta \boldsymbol{A}_1 \boldsymbol{B}_1^\top (\boldsymbol{B}_2^\top)^\dagger. \end{aligned} \tag{15}$$

Similarly, $\delta \boldsymbol{B}_2 = \delta \boldsymbol{B}_1 \boldsymbol{A}_1^\top (\boldsymbol{A}_2^\top)^\dagger$. Consequently, we have

$$\boldsymbol{A}_1 \boldsymbol{B}_1^\top + \delta \boldsymbol{A}_1 \boldsymbol{B}_1^\top + \boldsymbol{A}_1 \delta \boldsymbol{B}_1^\top + \delta \boldsymbol{A}_1 \delta \boldsymbol{B}_1^\top = \boldsymbol{A}_2 \boldsymbol{B}_2^\top + \delta \boldsymbol{A}_2 \boldsymbol{B}_2^\top + \boldsymbol{A}_2 \delta \boldsymbol{B}_2^\top + \delta \boldsymbol{A}_2 \delta \boldsymbol{B}_2^\top.$$

This demonstrates that our proposed method satisfies transformation invariance. Note that this simplified update rule does not yet incorporate accumulated first and second moments, which will be addressed in the following paragraphs.

**Incorporating second moment.** Adaptive optimizers typically employ accumulated second moments for preconditioning. A naive approach might involve replacing the $\bar{\nabla}\boldsymbol{A}^\top\bar{\nabla}\boldsymbol{A}$ term in (14) with its accumulated sum over training iterations $t \in \{1, 2, \dots, T\}$:

$$\sum\nolimits_{t=1}^{T}\bar{\nabla}\boldsymbol{A}_t{}^\top\bar{\nabla}\boldsymbol{A}_t.$$

However, since each $\bar{\nabla}\boldsymbol{A}_t$ is computed with respect to the basis at a specific step, directly summing them is mathematically incorrect. Instead, we must account for the varying basis at each step. To achieve this, we accumulate the second moment as follows:

$$\bar{\boldsymbol{V}}_{\boldsymbol{A}_t} = \boldsymbol{P}_{\boldsymbol{A}_t}\bar{\boldsymbol{V}}_{\boldsymbol{A}_{t-1}}\boldsymbol{P}_{\boldsymbol{A}_t}^\top + \bar{\nabla}\boldsymbol{A}_t{}^\top\bar{\nabla}\boldsymbol{A}_t, \tag{16}$$

where $\bar{\boldsymbol{V}}_{\boldsymbol{A}_{t-1}}$ is the accumulated second moment based on the previous basis at step $t-1$, and $\boldsymbol{P}_{\boldsymbol{A}_t} := (\boldsymbol{U}_{\boldsymbol{B}_t})^\top\boldsymbol{U}_{\boldsymbol{B}_{t-1}}$ transforms it to the new basis at step $t$. During the adjustment,

$$\mathrm{Tr}(\boldsymbol{P}_{\boldsymbol{A}_t}\bar{\boldsymbol{V}}_{\boldsymbol{A}_{t-1}}\boldsymbol{P}_{\boldsymbol{A}_t}^\top) \leq \mathrm{Tr}(\bar{\boldsymbol{V}}_{\boldsymbol{A}_{t-1}}),$$

indicating a potential loss of information from the accumulated second moment. To quantify this loss, for symmetric positive definite matrices $\boldsymbol{E}_1, \boldsymbol{E}_2 \in \mathbb{R}^{r\times r}$, we define

$$d_\lambda(\boldsymbol{E}_1, \boldsymbol{E}_2) \equiv \max_i |\lambda_i(\boldsymbol{E}_1) - \lambda_i(\boldsymbol{E}_2)| \leq \min_{\boldsymbol{U}}\|\boldsymbol{E}_1 - \boldsymbol{U}\boldsymbol{E}_2\boldsymbol{U}^\top\|,$$

where $\lambda_i(\boldsymbol{E})$ is the $i$-th eigenvalue of $\boldsymbol{E}$, and $\boldsymbol{U} \in \mathbb{R}^{r\times r}$ is an orthogonal matrix that reflects our freedom to choose the basis. We then define the "escaped mass" as

$$\rho_{\boldsymbol{A}_t} = \rho_{\boldsymbol{A}_{t-1}} + d_\lambda(\bar{\boldsymbol{V}}_{\boldsymbol{A}_{t-1}}, \boldsymbol{P}_{\boldsymbol{A}_t}\bar{\boldsymbol{V}}_{\boldsymbol{A}_{t-1}}\boldsymbol{P}_{\boldsymbol{A}_t}^\top). \tag{17}$$

To compensate for this, we add $\rho_{\boldsymbol{A}_t}\boldsymbol{I}$ to our preconditioner, ensuring that $\bar{\boldsymbol{V}}_{\boldsymbol{A}_t}+\rho_{\boldsymbol{A}_t}\boldsymbol{I}$ monotonically increases under a suitable choice of basis, even though the choice of basis does not influence the actual update.

Finally, our unmagnified preconditioned step, when incorporating second moment, can be written as

$$\bar{\boldsymbol{S}}_{\boldsymbol{A}_t} = \bar{\nabla}\boldsymbol{A}_t(\bar{\boldsymbol{V}}_{\boldsymbol{A}_t} + \rho_{\boldsymbol{A}_t}\boldsymbol{I})^{-1/2}. \tag{18}$$

Note that similar to Adam, we can turn (16) into the Exponential Moving Average (EMA) form, where we multiple the first term by $1-\beta_2$ and the second term by $\beta_2$, with the hyper-parameter $\beta_2 \in (0, 1)$ controls the decay rate.

**Incorporating first moment.** Similar to the second moment, the first moment must also be adjusted for changes in the basis using a projection matrix. The update rule for maintaining the first moment can then be written as $\bar{\boldsymbol{M}}_{\boldsymbol{A}_t} = \beta_1\bar{\boldsymbol{M}}_{\boldsymbol{A}_{t-1}}\boldsymbol{P}_{\boldsymbol{A}_t}^\top + (1-\beta_1)\bar{\boldsymbol{S}}_{\boldsymbol{A}_t}.$

Our final proposed update rule, incorporating both first and second moment, is

$$\delta\boldsymbol{A}_t = -\eta\bar{\boldsymbol{M}}_{\boldsymbol{A}_t}(\boldsymbol{R}_{\boldsymbol{B}}^\top)^\dagger. \tag{19}$$

---

**Algorithm 1 LoRA-RITE**

---

1: Initialize: unmagnified first and second moment $\bar{\boldsymbol{M}}_{\boldsymbol{A}_0} = \boldsymbol{0}, \bar{\boldsymbol{V}}_{\boldsymbol{A}_0} = \boldsymbol{0}$
2: **for** $t = 1\dots T$ **do**
3:     Compute the gradient $\nabla\boldsymbol{A}_t$;
4:     Compute polar decomposition of the LoRA factor $\boldsymbol{B}_t$: $\boldsymbol{B}_t = \boldsymbol{U}_{\boldsymbol{B}_t}\boldsymbol{R}_{\boldsymbol{B}_t}$;
5:     Compute the unmagnified gradient $\bar{\nabla}\boldsymbol{A}_t = \nabla\boldsymbol{A}_t\boldsymbol{R}_{\boldsymbol{B}_t}^\dagger$ and $\boldsymbol{P}_{\boldsymbol{A}_t} = (\boldsymbol{U}_{\boldsymbol{B}_t})^\top\boldsymbol{U}_{\boldsymbol{B}_{t-1}}$;
6:     Update the unmagnified second moment $\bar{\boldsymbol{V}}_{\boldsymbol{A}_t} = \boldsymbol{P}_{\boldsymbol{A}_t}\bar{\boldsymbol{V}}_{\boldsymbol{A}_{t-1}}\boldsymbol{P}_{\boldsymbol{A}_t}^\top + (\bar{\nabla}\boldsymbol{A}_t)^\top\bar{\nabla}\boldsymbol{A}_t/m$;
7:     Update the escaped mass $\rho_{\boldsymbol{A}_t} = \rho_{\boldsymbol{A}_{t-1}} + d_\lambda(\bar{\boldsymbol{V}}_{\boldsymbol{A}_{t-1}}, \boldsymbol{P}_{\boldsymbol{A}_t}\bar{\boldsymbol{V}}_{\boldsymbol{A}_{t-1}}\boldsymbol{P}_{\boldsymbol{A}_t}^\top)$;
8:     Compute the unmagnified precondition step $\bar{\boldsymbol{S}}_{\boldsymbol{A}_t} = \bar{\nabla}\boldsymbol{A}_t(\bar{\boldsymbol{V}}_{\boldsymbol{A}_t} + \rho_{\boldsymbol{A}_t}\boldsymbol{I})^{-1/2}$;
9:     Update the unmagnified first moment $\bar{\boldsymbol{M}}_{\boldsymbol{A}_t} = \beta_1\bar{\boldsymbol{M}}_{\boldsymbol{A}_{t-1}}\boldsymbol{P}_{\boldsymbol{A}_t}^\top + (1-\beta_1)\bar{\boldsymbol{S}}_{\boldsymbol{A}_t}$;
10:     Update model parameters $\boldsymbol{A}_{t+1} = \boldsymbol{A}_t - \eta_t\bar{\boldsymbol{M}}_{\boldsymbol{A}_t}(\boldsymbol{R}_{\boldsymbol{B}}^\top)^\dagger$.
11: **end for**

---

Our proposed algorithm, LoRA-RITE (**R**obust **I**nvariant **T**ransformation **E**quilibration for LoRA training), is summarized as Algorithm 1, where we show the updates for $\boldsymbol{A}$, and update for $\boldsymbol{B}$ can be derived in the same way. Note that we have shown that the main update rule of (18) satisfies transformation invariance, and this property can be extended even after adding the first and second moment into the algorithm, as shown in the following theorem (proof in Appendix).

**Theorem 2.** *In Algorithm 1, every unmagnified term is consistent across all equivalent LoRA pairs. Consequently, Algorithm 1 is transformation invariant.*

**Time and Space Complexity.** The time and space complexity of our algorithm is similar to first order methods like Adam when $r \ll m, n$. In each iteration of Algorithm 1, the dominant computational costs arise from (1) polar-decomposition for $m$-by-$r$ and $n$-by-$r$ matrices which takes $\mathcal{O}(nr^2 + mr^2)$ time, (2) matrix inverses and roots for $r$-by-$r$ matrices which takes $\mathcal{O}(r^3)$ time, and (3) matmuls with time complexity $\mathcal{O}(nr^2 + mr^2)$. Thus, the overall complexity per step is $\mathcal{O}(mr^2 + nr^2 + r^3)$. It is only $r$ times slower than Adam, and since $r$ is very small, this overhead is negligible when comparing with the back-propagating time. The memory cost of our method is $\mathcal{O}(mr + nr)$ which is the same as Adam. We summarize the time and space complexity of our method versus some commonly used optimizers in Table 7 in the Appendix.

## 3.3 THEORETICAL ANALYSIS

Following previous work (Gupta et al., 2018; Feinberg et al., 2023), we provide a convergence analysis of the proposed algorithm within the online optimization framework (Hazan et al., 2016; Shalev-Shwartz et al., 2012). In online convex optimization setting, a parameter $\boldsymbol{\theta}_t \in \mathcal{K}$ is chosen iteratively, where $\mathcal{K}$ is a convex decision set. After each decision- $\boldsymbol{\theta}_t$, a convex loss function $f_t$ is revealed, potentially chosen adversarially. The regret accumulated by the algorithm up to step $T$ is defined as

$$\text{Regret}_T = \sum_{t=1}^T f_t(\boldsymbol{\theta}_t) - \min_{\boldsymbol{\theta} \in \mathcal{K}} \sum_{t=1}^T f_t(\boldsymbol{\theta}).$$

In the online convex optimization analysis, we bound the first-order term $\nabla\boldsymbol{\theta}_t^\top(\boldsymbol{\theta}_t - \boldsymbol{\theta}^*)$ where $\boldsymbol{\theta}^*$ represents an arbitrary minimizer, and then use convexity of $f_t$ to connect it to the loss function. However, due to the inherent structure of LoRA, loss functions $f_t$, $t \in \{1, \ldots, T\}$ are not convex with respect to $\boldsymbol{\theta}$. Therefore, we directly bound the first-order term instead.

We assume for the fine-tuned weight $\boldsymbol{Z}$ of each layer, the convex decision set imposes the following constrains: $\|\boldsymbol{A}\|_F \leq D_{\boldsymbol{A}}, \|\boldsymbol{B}\|_F \leq D_{\boldsymbol{B}}$, where $\|\cdot\|$ denotes the Frobenius norm. Additionally, we assume the gradient satisfies $\|\nabla\boldsymbol{Z}\|_F \leq G$. Following previous work, we analyze convergence in the simplified scenario where the first moment is omitted and the second moment is a summation, similar to Adagrad. For LoRA-RITE, our theoretical analysis yields the following result:

**Theorem 3.** *LoRA-RITE satisfies:*

$$\frac{1}{T}\sum_{t=1}^T \frac{1}{\eta}\nabla\boldsymbol{\theta}_t^\top(\boldsymbol{\theta}_t - \boldsymbol{\theta}_{t+1}) = \mathcal{O}(GT^{-1/2}),$$

*where $\eta$ is a fixed constant learning rate.*

This theorem shows that the method either converges to a particular stable solution or just move around in directions that does not change the function value, suggesting a form of convergence. To further strengthen the guarantee, we introduce an additional assumption:

**Assumption 1.** *Let $\bar{\boldsymbol{X}}_{\boldsymbol{A}_t} = (\bar{\boldsymbol{V}}_{\boldsymbol{A}_t} + \rho_{\boldsymbol{A}_t}\boldsymbol{I})^{-1/2}$ be the unmagnified preconditioner $\boldsymbol{P}_{\boldsymbol{A}_t} = (\boldsymbol{U}_{\boldsymbol{B}_t})^\top\boldsymbol{U}_{\boldsymbol{B}_{t-1}}$, and $\boldsymbol{Q}_{\boldsymbol{A}_t} = (\boldsymbol{R}_{\boldsymbol{B}_t}^\top)^\dagger\boldsymbol{R}_{\boldsymbol{B}_{t-1}}^\top$, then we have*

$$\|\bar{\boldsymbol{X}}_{\boldsymbol{A}_t}^{-1} - \boldsymbol{Q}_{\boldsymbol{A}_t}\bar{\boldsymbol{X}}_{\boldsymbol{A}_{t-1}}^{-1}\boldsymbol{Q}_{\boldsymbol{A}_t}^T\| \leq \mu\|\bar{\boldsymbol{X}}_{\boldsymbol{A}_t}^{-1} - \boldsymbol{P}_{\boldsymbol{A}_t}\bar{\boldsymbol{X}}_{\boldsymbol{A}_{t-1}}^{-1}\boldsymbol{P}_{\boldsymbol{A}_t}^T\|$$

*and*

$$\text{Tr}(\boldsymbol{P}_{\boldsymbol{A}_t}\bar{\boldsymbol{X}}_{\boldsymbol{A}_{t-1}}^{-1}\boldsymbol{P}_{\boldsymbol{A}_t}^T) \geq \text{Tr}(\bar{\boldsymbol{X}}_{\boldsymbol{A}_{t-1}}^{-1} + c(\rho_{\boldsymbol{A}_{t-1}}^{1/2} - \rho_{\boldsymbol{A}_t}^{1/2})\boldsymbol{I}).$$

This assumption essentially constrains the change in $\boldsymbol{U}_{\boldsymbol{B}_t}$ and $\boldsymbol{R}_{\boldsymbol{B}_t}$ to be relatively smooth. Under this assumption, we can establish the following stronger convergence result:

**Theorem 4.** *Under Assumption 1, our proposed method satisfies:*

$$\frac{1}{T}\sum_{t=1}^T \nabla\boldsymbol{\theta}_t^\top(\boldsymbol{\theta}_t - \boldsymbol{\theta}^*) = O(GD_{\boldsymbol{A}}D_{\boldsymbol{B}}T^{-1/2}).$$

Our analysis closely resembles that of one-sided matrix Adagrad. The key idea is to have a change of variable for both $\boldsymbol{A}$ and $\boldsymbol{B}$ such that all the quantities get replace by its unmagnified counterparts.

Compared to one-sided matrix Adagrad, which has a regret bound of

$$\mathcal{O}(G(D_{\boldsymbol{A}}^2 + D_{\boldsymbol{B}}^2)T^{-1/2}) \text{ which is higher than } \mathcal{O}(GD_{\boldsymbol{A}}D_{\boldsymbol{B}}T^{-1/2}),$$

the regret bound of LoRA-RITE. The above inequality is using $D_{\boldsymbol{A}}^2 + D_{\boldsymbol{B}}^2 \geq 2D_{\boldsymbol{A}}D_{\boldsymbol{B}}$, where the difference between both sides of inequality is large when the two LoRA factors exhibit imbalance in magnitudes - $D_{\boldsymbol{A}}$ and $D_{\boldsymbol{B}}$. This advantage is particularly relevant because previous work has shown that LoRA factors often exhibit such imbalances (Hayou et al., 2024), which can also be seen in Figure 1, providing an explanation for the strong empirical performance of our method.

## 4 RELATED WORK

**Related Optimizers.** Adaptive first-order optimizers like Adagrad (Duchi et al., 2011) utilize accumulated second moments, essentially diagonal preconditioners, to scale updates for each coordinate. This approach, adopted by optimizers like Adam (Kingma & Ba, 2014) and RMSProp (Tieleman & Hinton, 2012), has become standard for training deep neural networks, including LoRA, and many other similar first-order methods have also been developed in the literature (Loshchilov & Hutter, 2017; Chen et al., 2024). However, as discussed in Section 3.1, these methods lack transformation invariance when applied to LoRA.

Several higher-order preconditioners have shown promise in various training scenarios (Shi et al., 2023). For example, Shampoo (Gupta et al., 2018) approximates the full second moment matrix using a Kronecker product, leading to the following preconditioned gradient:

$$L^{-1/4}GR^{-1/4}, \;\; L = L + GG^\top, \;\; R = R + G^\top G, \tag{20}$$

where $L \in \mathbb{R}^{m \times m}, R \in \mathbb{R}^{n \times n}$ are the left and right preconditioner matrices, and $G \in \mathbb{R}^{m \times n}$ is the gradient. Many other higher-order methods follow this framework (Martens & Grosse, 2015; Morwani et al., 2024; Duvvuri et al., 2024). These methods incur $\mathcal{O}(m^2 + n^2)$ additional memory overhead and require periodic computation of roots of $L$ and $R$ with $\mathcal{O}(m^3 + n^3)$ computational cost. This complexity significantly exceeds that of our proposed method, as demonstrated in Table 7. Comparing (20) and (18) reveals that our method applies preconditioning only to the low-rank side of LoRA, resulting in negligible overhead. Furthermore, unlike our provably transformation-invariant approach, Shampoo-based methods lack this property.

LARS (You et al., 2017) and Lamb (You et al., 2020) are layer-wise adaptive optimization methods originally designed for large batch training. They dynamically adjust the update norm for each weight matrix based on its current norm, which ensures scalar scale invariance. Nonetheless, they still lack transformation invariance.

**Variants of LoRA.** As large language models (LLMs) grow in size, full fine-tuning on downstream tasks becomes increasingly resource-intensive. Parameter-efficient fine-tuning (PEFT) methods such as (Houlsby et al., 2019; He et al., 2022b;a; Lester et al., 2021; Li & Liang, 2021) have emerged to address this issue by reducing the number of trainable paramters. As a popular PEFT algorithm, LoRA (Hu et al., 2022) has been the subject of extensive research, with numerous variations and improvements proposed. One line of research focuses on dynamically adjusting the LoRA rank during training. This includes DyLoRA (Valipour et al., 2023), IncreLoRA (Zhang et al., 2023a), and AdaLoRA (Zhang et al., 2023b). Another approach involves enhancing LoRA performance through the addition of extra scaling matrices, which includes DoRA (Liu et al., 2024) and DeepLoRA (Yaras et al., 2024). These directions are orthogonal to our work.

Regarding LoRA optimization, Hayou et al. (2024) highlight the limitations of traditional optimizers as they fail to achieve efficient feature learning. To address this issue, they propose LoRA+, which uses two different learning rates $\eta_A$ and $\eta_B$ for LoRA weights. However, this leads to an extra hyperparameter to be tuned in practice. In contrast, Zhang & Pilanci (2024) propose the use of matrix preconditioning methods to achieve efficient feature learning. They propose the use of Riemannian gradient descent for LoRA optimization. As far as we know, Riemannian gradient descent is the only method in the literature that satisfies transformation invariance. However, similar to gradient descent, Riemannian gradient descent does not incorporate momentum and adaptivity, so it performs worse than Adam in their experiments. To improve the performance, they propose to combine Riemannian gradient descent with element-wise Adam, which becomes ScaledAdam. However, this combination makes ScaledAdam no longer transformation invariant.

## 5 EXPERIMENTAL RESULTS

We evaluate the proposed LoRA optimizer against other optimizers across a range of datasets. This includes the Super-Natural Instructions dataset, a comprehensive collection of diverse NLP tasks, as well as four standard LLM benchmarking datasets.

We compare the following optimizers:

- Adam (Kingma & Ba, 2014): The most widely used default optimizer for LoRA finetuning.

Table 1: Experimental results on the Super-Natural instruction dataset.

| Model | Optimizer | Cause Effect Classification | Coreference Resolution | Title Generation | Data to Text | Global |
|-------|-----------|------------------|--------------|--------------|--------------|--------------|
| Gemma-2B | Adam | 58.93 | 77.06 | 51.30 | 55.52 | 50.51/74.54 |
| | LoRA+ | 58.84 | 76.08 | 51.32 | 55.68 | 49.76/74.20 |
| | ScaledAdam | 58.71 | 77.55 | 51.16 | 55.69 | 49.40/74.01 |
| | Shampoo | 58.11 | 77.17 | 51.30 | 55.48 | 50.79/74.74 |
| | Lamb | 60.97 | 80.69 | 52.26 | 55.85 | 53.53/76.43 |
| | LoRA-RITE | **61.26** | **82.02** | **52.26** | **55.98** | **55.11/77.12** |
| Gemma-7B | Adam | 67.17 | 86.05 | 51.58 | 55.38 | 58.46/78.17 |
| | LoRA+ | 65.50 | 86.67 | 51.51 | 55.34 | 58.19/78.29 |
| | ScaledAdam | 65.79 | 85.05 | 51.61 | 55.40 | 57.32/77.92 |
| | Shampoo | 66.29 | 85.62 | 51.86 | 55.43 | 57.99/78.27 |
| | Lamb | 69.62 | 86.57 | 51.87 | 55.5 | 57.79/78.18 |
| | LoRA-RITE | **71.26** | **88.14** | **52.17** | **55.62** | **59.71/79.05** |

Table 2: Experimental results on LLM benchmarking datasets.

| Model | Optimizer | HellaSwag | ArcChallenge | GSM8K | OpenBookQA | Avg. |
|-------|-----------|-----------|--------------|-------|------------|------|
| Gemma-2B | Adam | 83.76 | 45.31 | 24.26 | 64.0 | 54.33 |
| | LoRA+ | 83.75 | 45.31 | 23.65 | 64.4 | 54.28 |
| | ScaledAdam | 83.52 | 45.22 | 23.96 | 64.8 | 54.38 |
| | Shampoo | 83.26 | 44.88 | 23.35 | 63.6 | 53.77 |
| | Lamb | 86.60 | 47.35 | 26.76 | 68.0 | 57.18 |
| | LoRA-RITE | **87.28** | **49.06** | **30.10** | **68.8** | **58.81** |
| Gemma-7B | Adam | 94.07 | 54.78 | 48.37 | 77.60 | 68.71 |
| | LoRA+ | 93.99 | 54.01 | 48.75 | 77.60 | 68.59 |
| | ScaledAdam | 93.31 | 52.90 | 48.07 | 75.80 | 67.52 |
| | Shampoo | 94.15 | 52.47 | 49.05 | 76.80 | 68.12 |
| | Lamb | 95.11 | 69.80 | 50.64 | 83.20 | 74.69 |
| | LoRA-RITE | **95.59** | **71.76** | **55.50** | **84.80** | **76.91** |

- LoRA+ (Hayou et al., 2024): Adam with different learning rates for $A$ and $B$. We set the learning rate of $B$ to be 4 times large than $A$, which is the value they used for decoder models.
- ScaledAdam (Zhang & Pilanci, 2024): A variant of Adam designed for LoRA optimization.
- Shampoo (Gupta et al., 2018): A well-known adaptive matrix preconditioning method. To obtain similar training time as the other methods, the block size is set to 512 and the preconditioners are updated every 100 steps.
- Lamb (You et al., 2020): A variant of Adam that normalizes the updates for each layer based on the norm of the parameters.
- LoRA-RITE: Our proposed optimizer that is transformation invariant.

For each optimizer applied on each data set, we search for the best learning rate from $2 * 10^{-6}$ to $2 * 10^{-2}$. The other hyperparameters are listed in the Appendix. For most of the experiments we chose rank $r = 16$ for LoRA, based on the ablation study over the rank. We conduct experiments on Gemma (Gemma Team et al., 2024) 2B, 7B, and mT5-XXL (Xue et al., 2021) using TPUs.

**Results on Super-Natural Instruction Dataset.** The Super-Natural instruction dataset (Wang et al., 2022) contains a collection of 1600+ NLP tasks, including both classification and generation tasks. We use a 10% split of the data for validation. Following (Wang et al., 2022), we use the exact match accuracy to evaluate classification and ROUGE-L score to evaluate generation tasks.

Table 1 presents the performance of individual fine-tuning on two classification and two generation tasks for 2,000 steps. It also includes the performance of fine-tuning on the global training set of over 1,600 tasks for 10,000 steps, reporting both exact match accuracy and ROUGE-L score evaluated on the global validation set. As shown in Table 1, our proposed method demonstrates superior performance across both classification and generation tasks. Compared to Adam, our method

Table 3: Ablation study on different ranks and different model architectures.

|  | Gemma-2B (rank=4) | Gemma-2B (rank=16) | mT5-XXL (rank=4) | mT5-XXL (rank=16) |
|---|---|---|---|---|
| Adam | 63.00 | 64.0 | 72.00 | 72.20 |
| ScaledAdam | 63.00 | 64.8 | 70.80 | 74.60 |
| Lamb | 67.80 | 68.0 | 70.40 | 73.40 |
| LoRA-RITE | **70.40** | **68.8** | **74.80** | **75.00** |

Table 4: Number of training steps per second for different optimizers. Shampoo preconditioner is updated once every 100 steps. LoRA-RITE has small overhead compared with first-order methods.

|  | Adam | LoRA+ | ScaledAdam | Shampoo | Lamb | LoRA-RITE |
|---|---|---|---|---|---|---|
| Gemma-2B | 0.948 | 0.930 | 0.917 | 0.837 | 0.929 | 0.878 |
| Gemma-7B | 0.120 | 0.120 | 0.114 | 0.112 | 0.116 | 0.114 |

achieves 2.3% to 4.9% accuracy improvements on the classification tasks and also shows significant improvements in the global training setting. Furthermore, we found that Lamb performs well on some of the data sets but there's still a significant gap between Lamb and LoRA-RITE. Since Lamb enforces scalar scale invariance but not transformation invariance, this result implicitly suggests that transformation invariance is crucial for achieving optimal performance.

**Results on other LLM Benchmarking Datasets.** We also evaluate the performance on common LLM benchmarking datasets, including HellaSwag (Zellers et al., 2019), ArcChallenge (Clark et al., 2018), GSM8K (Cobbe et al., 2021), and OpenBookQA (Mihaylov et al., 2018). The summary information of these datasets is in the Appendix. The results are presented in Table 2. We can observe that the trend is similar to the SuperNatural instruction results, where LoRA-RITE achieves the best performance on all the data sets, and Lamb is usually the second best optimizer.

**Ablation Study.** We conduct an ablation study on the choice of different LoRA ranks and model architectures. Specifically, we considered rank 4 and 16 on both Gemma 2B (decoder only) and mT5-XXL (encoder-decoder) on the OpenBookQA dataset. As we can see from Table 3, our proposed method performs consistently well across different LoRA ranks. Furthermore, our method can be successfully applied to mT5-XXL which has an encoder-decoder architecture, showing the generalizability of the proposed optimizer.

**Training Speed Comparison.** We compare the training speed of different optimizers. Table 4 shows the number of training steps per second for different optimizers with LoRA rank 16 on the OpenBookQA dataset using TPUv5e. As we can see, LoRA-RITE is only 8% slower than Adam on Gemma 2B, while the difference decreases to 5% when model size increases to 7B. Also, Shampoo is slower than LoRA-RITE in this case despite the fact that it recomputes the preconditioner with much lower frequency (once every 100 steps). This is due to our approach of preconditioning only the low-rank side of the LoRA factors.

## 6 CONCLUSIONS

Current LoRA optimization techniques lack transformation invariance, which implies that equivalent LoRA parameterizations can yield significantly different updates. This hinders efficient feature learning and often leads to suboptimal solutions in practice. We introduce a novel, transformation-invariant optimization algorithm with comparable time and memory overhead to Adam. Our algorithm consistently achieves higher accuracy than existing LoRA optimizers across diverse datasets and models.

**Limitations.** Although this work introduces a better optimizer for LoRA, it is important to acknowledge that LoRA itself has limitations. For instance, LoRA has smaller representational power and may result in a minor performance decrease compared to full fine-tuning. Also, how to select rank to strike a good trade-off between efficiency and accuracy may be non-trivial in practice. The work focuses on addressing transformation-invariance when the optimization problem can be written in the form of $f(AB^\top)$, and this assumption may not hold for other parameter-efficient structures beyond LoRA. Applying LoRA-RITE to ensure transformation invariance for the other more complicated LoRA variants will be an interesting future direction.

ACKNOWLEDGMENTS

We thank Sashank Reddi and Corinna Cortes for useful feedback. The code for our project is available at `https://github.com/gkevinyen5418/LoRA-RITE`.

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

# A APPENDIX

## A.1 HYPERPARAMETERS

Table 5 shows our hyperparameters. We set weight decay and dropout probability to $0$ as our early experiments suggest that setting a non-zero value does not improve the performance of the baselines.

Table 5: The setting for hyperparameters.

| Hyperparameter | Value |
|---|---|
| Learning rate | $2 * 10^{-6}$ to $2 * 10^{-2}$ |
| Weight decay | 0 |
| Dropout prob | 0 |
| LoRA target | $q_{proj}$, $k_{proj}$, $v_{proj}$, $o_{proj}$ |
| LoRA rank | 16 |
| LoRA $\alpha$ | 16 |
| Batch size | 16 |
| Train step | 2000 |
| LR schedule | Linear decay |
| Warmup step | 100 |
| Evaluation period | 100 |
| Momentum $\beta_1$ | 0.9 |
| Second moment $\beta_2$ | 0.999 |

## A.2 DATASETS

Table 6 shows the summary information of the LLM benchmarking datasets. We use the test set to evaluate ArcChallenge, as it is much larger than the development set.

Table 6: Summary information of the LLM benchmarking datasets.

| Dataset | #Train | #Dev | #Test | Split for Eval |
|---|---|---|---|---|
| HellaSwag | 39905 | 10042 | 10003 | Dev |
| ArcChallenge | 1119 | 299 | 1172 | Test |
| GSM8K | 7473 | NA | 1319 | Test |
| OpenBookQA | 4957 | 500 | 500 | Dev |

## A.3 PROOF OF THEOREM 1

Let $\|\boldsymbol{A}_1\| = \boldsymbol{\theta}(n^a)$, $\|\boldsymbol{B}_1\| = \boldsymbol{\theta}(n^b)$, $\|\nabla \boldsymbol{Z}\| = \boldsymbol{\theta}(n^c)$, $\eta = \boldsymbol{\theta}(n^d)$, where $\eta$ is the learning rate and $n$ is the network width. Since $\boldsymbol{Z} = \boldsymbol{A}_1 \boldsymbol{B}_1^T$, from chain rule we know $\nabla \boldsymbol{A} = \nabla \boldsymbol{Z} \boldsymbol{B}$ and $\nabla \boldsymbol{B} = \nabla \boldsymbol{Z}^T \boldsymbol{A}$. Since the update rule is symmetric, we can express the updates as

$$\|\delta \boldsymbol{A}_1\| = \boldsymbol{\theta}(n^{xa+yb+zc+d}), \|\delta \boldsymbol{B}_1\| = \boldsymbol{\theta}(n^{xb+ya+zc+d}).$$

If the update rule is scalar scale invariant, then for any $\boldsymbol{A}_2 = n^\delta \boldsymbol{A}_1$, $\boldsymbol{B}_2 = n^{-\delta} \boldsymbol{B}_1$ we have

$$\|\delta \boldsymbol{A}_1\| \|\boldsymbol{B}_1\| = \|\delta \boldsymbol{A}_2\| \|\boldsymbol{B}_2\|,$$

which means

$$xa + (y+1)b + zc + d = x(a+\delta) + (y+1)(b-\delta) + zc + d,$$

thus $x\delta - (y+1)\delta = 0$ for all $\delta$, which means $y = x - 1$. Consequently, we have

$$\|\delta \boldsymbol{A}_1\| \|\boldsymbol{B}_1\| = \boldsymbol{\theta}(n^{xa+(y+1)b+sc+d}) = \boldsymbol{\theta}(n^{xa+xb+sc+d}).$$

Table 7: Time and space complexity comparison for LoRA optimization.

| Algorithm | Time Complexity | Space Complexity |
|---|---|---|
| Forward/Backward | $\Omega(nm)$ | $\Omega(nm)$ |
| Full Matrix Adagrad (Duchi et al., 2011) | $O(m^3r^3 + n^3r^3)$ | $O(m^2r^2 + n^2r^2)$ |
| Adam (Kingma & Ba, 2014) | $O(mr + nr)$ | $O(mr + nr)$ |
| Lamb (You et al., 2020) | $O(mr + nr)$ | $O(mr + nr)$ |
| Shampoo (Gupta et al., 2018) | $O(m^3 + n^3 + r^3)$ | $O(m^2 + n^2 + r^2)$ |
| KFAC (Martens & Grosse, 2015) | $O(m^3 + n^3 + r^3)$ | $O(m^2 + n^2 + r^2)$ |
| ScaledAdam (Zhang & Pilanci, 2024) | $O(mr^2 + nr^2)$ | $O(mr + nr)$ |
| LoRA-RITE (our proposed) | $O(mr^2 + nr^2)$ | $O(mr + nr + r^2)$ |

Similarly, we have

$$\|\boldsymbol{A}_1\|\|\delta\boldsymbol{B}_1\| = \boldsymbol{\theta}(n^{xb+(y+1)a+sc+d}) = \boldsymbol{\theta}(n^{xb+xa+sc+d}).$$

Since these two are equal, we can achieve efficient feature learning

$$\|\boldsymbol{A}\|\|\delta\boldsymbol{B}\|\|\boldsymbol{x}\| = \|\delta\boldsymbol{A}\|\|\boldsymbol{B}\|\|\boldsymbol{x}\| = \boldsymbol{\theta}(1),$$

where $\boldsymbol{x}$ is the input vector, by selecting a proper learning rate $\eta$.

### A.4 PROOF OF THEOREM 2

In Algorithm 1, for two equivalent LoRA pairs $(\boldsymbol{A}_1, \boldsymbol{B}_1), (\boldsymbol{A}_2, \boldsymbol{B}_2)$, their obtained basis $\boldsymbol{U}_{\boldsymbol{B}_1}$ and $\boldsymbol{U}_{\boldsymbol{B}_2}$ could be different.

Consequently, we introduce the concept of consistency, which means a matrix output of $f(\boldsymbol{A}, \boldsymbol{B})$ is the same across all equivalent LoRA pairs, up to the choice of the basis.

**Definition 3** (Consistency). *For matrix $\boldsymbol{X_A} \in \mathbb{R}^{m \times r}$, we call it consistent if*

$$\boldsymbol{X_{A_1}}\boldsymbol{U}_{\boldsymbol{B}_1}^\top = \boldsymbol{X_{A_2}}\boldsymbol{U}_{\boldsymbol{B}_2}^\top \in \mathbb{R}^{m \times n}$$

*for all equivalent LoRA pairs $(\boldsymbol{A}_1, \boldsymbol{B}_1), (\boldsymbol{A}_2, \boldsymbol{B}_2)$. Similarly, for $\boldsymbol{H_A} \in \mathbb{R}^{r \times r}$, we call it consistent if*

$$\boldsymbol{U}_{\boldsymbol{B}_1}\boldsymbol{H_{A_1}}\boldsymbol{U}_{\boldsymbol{B}_1}^\top = \boldsymbol{U}_{\boldsymbol{B}_2}\boldsymbol{H_{A_2}}\boldsymbol{U}_{\boldsymbol{B}_2}^\top \in \mathbb{R}^{n \times n}$$

*for all equivalent LoRA pairs $(\boldsymbol{A}_1, \boldsymbol{B}_1), (\boldsymbol{A}_2, \boldsymbol{B}_2)$.*

Additionally, to simplify the notation, from now on we use $\boldsymbol{A}, \boldsymbol{B}$ to stand for $\boldsymbol{A}_1, \boldsymbol{B}_1$ and use $\hat{\boldsymbol{A}}, \hat{\boldsymbol{B}}$ to stand for $\boldsymbol{A}_2, \boldsymbol{B}_2$.

Then, we proceed to prove Theorem 2. First, one should note the fact that

$$\boldsymbol{U}_{\boldsymbol{B}}\boldsymbol{U}_{\boldsymbol{B}}^\top = \boldsymbol{U}_{\hat{\boldsymbol{B}}}\boldsymbol{U}_{\hat{\boldsymbol{B}}}^\top$$

for all equivalent LoRA pairs. Consequently,

$$\begin{aligned}
\boldsymbol{U}_{\boldsymbol{B}}(\bar{\nabla}\boldsymbol{A})^\top\bar{\nabla}\boldsymbol{A}\boldsymbol{U}_{\boldsymbol{B}}^\top &= \boldsymbol{U}_{\boldsymbol{B}}\boldsymbol{U}_{\boldsymbol{B}}^\top\nabla\boldsymbol{Z}^\top\nabla\boldsymbol{Z}\boldsymbol{U}_{\boldsymbol{B}}\boldsymbol{U}_{\boldsymbol{B}}^\top \\
&= \boldsymbol{U}_{\hat{\boldsymbol{B}}}\boldsymbol{U}_{\hat{\boldsymbol{B}}}^\top\nabla\boldsymbol{Z}^\top\nabla\boldsymbol{Z}\boldsymbol{U}_{\hat{\boldsymbol{B}}}\boldsymbol{U}_{\hat{\boldsymbol{B}}}^\top = \boldsymbol{U}_{\hat{\boldsymbol{B}}}(\bar{\nabla}\boldsymbol{A})^\top\bar{\nabla}\boldsymbol{A}\boldsymbol{U}_{\hat{\boldsymbol{B}}}^\top,
\end{aligned} \tag{21}$$

which shows that $(\bar{\nabla}\boldsymbol{A})^\top\bar{\nabla}\boldsymbol{A}$ is consistent.

Second, we can prove $\bar{\boldsymbol{V}}_{\boldsymbol{A}_t}$ is consistent by mathematical induction. The base case $\bar{\boldsymbol{V}}_{\boldsymbol{A}_0} = \boldsymbol{0}$ is consistent. Assuming $\bar{\boldsymbol{V}}_{\boldsymbol{A}_{t-1}}$ is consistent, then

$$\begin{aligned}
&\boldsymbol{U}_{\boldsymbol{B}_{t-1}}\boldsymbol{P}_{\boldsymbol{A}_t}\bar{\boldsymbol{V}}_{\boldsymbol{A}_{t-1}}\boldsymbol{P}_{\boldsymbol{A}_t}^\top(\boldsymbol{U}_{\boldsymbol{B}_t})^\top \\
=&\boldsymbol{U}_{\boldsymbol{B}_{t-1}}(\boldsymbol{U}_{\boldsymbol{B}_t})^\top\boldsymbol{U}_{\boldsymbol{B}_{t-1}}\bar{\boldsymbol{V}}_{\boldsymbol{A}_{t-1}}\boldsymbol{U}_{\boldsymbol{B}_{t-1}}(\boldsymbol{U}_{\boldsymbol{B}_t})^\top(\boldsymbol{U}_{\boldsymbol{B}_t})^\top \\
=&\boldsymbol{U}_{\hat{\boldsymbol{B}}_{t-1}}(\boldsymbol{U}_{\hat{\boldsymbol{B}}_t})^\top\boldsymbol{U}_{\hat{\boldsymbol{B}}_{t-1}}\bar{\boldsymbol{V}}_{\hat{\boldsymbol{A}}_{t-1}}\boldsymbol{U}_{\hat{\boldsymbol{B}}_{t-1}}(\boldsymbol{U}_{\hat{\boldsymbol{B}}_t})^\top(\boldsymbol{U}_{\hat{\boldsymbol{B}}_t})^\top, \\
=&\boldsymbol{U}_{\hat{\boldsymbol{B}}_{t-1}}\boldsymbol{P}_{\hat{\boldsymbol{A}}_t}\bar{\boldsymbol{V}}_{\hat{\boldsymbol{A}}_{t-1}}\boldsymbol{P}_{\hat{\boldsymbol{A}}_t}^\top(\boldsymbol{U}_{\hat{\boldsymbol{B}}_t})^\top
\end{aligned} \tag{22}$$

which means $\boldsymbol{P}_{\boldsymbol{A}_t}\bar{\boldsymbol{V}}_{\boldsymbol{A}_{t-1}}\boldsymbol{P}_{\boldsymbol{A}_t}^\top$ and thus

$$\bar{\boldsymbol{V}}_{\boldsymbol{A}_t} = \boldsymbol{P}_{\boldsymbol{A}_t}\bar{\boldsymbol{V}}_{\boldsymbol{A}_{t-1}}\boldsymbol{P}_{\boldsymbol{A}_t}^\top + \bar{\nabla}\boldsymbol{A}_t^\top\bar{\nabla}\boldsymbol{A}_t$$

is consistent.

Lastly, since

$$\boldsymbol{U}_B(\bar{\boldsymbol{V}}_{\boldsymbol{A}_t} + \rho_{\boldsymbol{A}_t}\boldsymbol{I})^{-1/2}\boldsymbol{U}_B^\top = (\boldsymbol{U}_B\bar{\boldsymbol{V}}_{\boldsymbol{A}_t}\boldsymbol{U}_B^\top + \rho_{\boldsymbol{A}_t}\boldsymbol{U}_B\boldsymbol{U}_B^\top)^{-1/2},$$

we have

$$\begin{aligned}
\bar{\boldsymbol{S}}_{\boldsymbol{A}_t}\boldsymbol{U}_{\boldsymbol{B}_t}^\top &= \bar{\nabla}\boldsymbol{A}_t(\bar{\boldsymbol{V}}_{\boldsymbol{A}_t} + \rho_{\boldsymbol{A}_t}\boldsymbol{I})^{-1/2}\boldsymbol{U}_{\boldsymbol{B}_t}^\top \\
&= \bar{\nabla}\boldsymbol{Z}_t\boldsymbol{U}_{\boldsymbol{B}_t}(\bar{\boldsymbol{V}}_{\boldsymbol{A}_t} + \rho_{\boldsymbol{A}_t}\boldsymbol{I})^{-1/2}\boldsymbol{U}_{\boldsymbol{B}_t}^\top \\
&= \bar{\nabla}\boldsymbol{Z}_t(\boldsymbol{U}_{\boldsymbol{B}_t}\bar{\boldsymbol{V}}_{\boldsymbol{A}_t}\boldsymbol{U}_{\boldsymbol{B}_t}^\top + \rho_{\boldsymbol{A}_t}\boldsymbol{U}_{\boldsymbol{B}_t}\boldsymbol{U}_{\boldsymbol{B}_t}^\top)^{-1/2} \\
&= \bar{\nabla}\boldsymbol{Z}_t(\boldsymbol{U}_{\hat{\boldsymbol{B}}_t}\bar{\boldsymbol{V}}_{\hat{\boldsymbol{A}}_t}\boldsymbol{U}_{\hat{\boldsymbol{B}}_t}^\top + \rho_{\hat{\boldsymbol{A}}_t}\boldsymbol{U}_{\hat{\boldsymbol{B}}_t}\boldsymbol{U}_{\hat{\boldsymbol{B}}_t}^\top)^{-1/2} \\
&= \bar{\boldsymbol{S}}_{\hat{\boldsymbol{A}}_t}\boldsymbol{U}_{\hat{\boldsymbol{B}}_t}^\top.
\end{aligned}$$

Thus, we can similarly proof by induction that both $\bar{\boldsymbol{S}}_{\boldsymbol{A}_t}$ and $\bar{\boldsymbol{M}}_{\boldsymbol{A}_t}$ are consistent, which completes our proof.

### A.5  PROOF OF THEOREM 3

For convenience, for matrix $\boldsymbol{X} \in \mathbb{R}^{m\times r}$, $\boldsymbol{H} \in \mathbb{R}^{r\times r}$, we define
$$\|\boldsymbol{X}\|_{\boldsymbol{H}} = \operatorname{Tr}(\boldsymbol{X}\boldsymbol{H}\boldsymbol{X}^\top)^{1/2}.$$

We also utilize the following lemma for online optimization.

**Lemma 1** (Lemma 5.13 Hazan et al. (2016)). *For online optimization, if $\boldsymbol{\theta}_t$ is updated as $\boldsymbol{\theta}_{t+1} = \boldsymbol{\theta}_t - \eta\boldsymbol{X}_t\boldsymbol{g}_t$, then we have*

$$\sum_{t=1}^T \nabla\boldsymbol{\theta}_t^\top(\boldsymbol{\theta}_t - \boldsymbol{\theta}^*) \leq \frac{1}{2\eta}\|\boldsymbol{\theta}_1 - \boldsymbol{\theta}_*\|_{\boldsymbol{X}_1^{-1}}^2 + \frac{\eta}{2}\sum_{t=1}^T(\boldsymbol{g}_t)^\top\boldsymbol{X}_t\boldsymbol{g}_t$$
$$+ \frac{1}{2\eta}\sum_{t=2}^T(\boldsymbol{\theta}_t - \boldsymbol{\theta}_*)^\top(\boldsymbol{X}_t^{-1} - \boldsymbol{X}_{t-1}^{-1})(\boldsymbol{\theta}_t - \boldsymbol{\theta}_*).$$

**Lemma 2** (Lemma 5.13, 5.14 Hazan et al. (2016)). *For arbitrary matrix $\boldsymbol{G}_t \in \mathbb{R}^{m\times r}$, $\boldsymbol{H}_t = \sum_{i=1}^t \boldsymbol{G}_i^\top\boldsymbol{G}_i$, we have*

$$\sum_{t=1}^T \|\boldsymbol{G}_t\|_{\boldsymbol{H}_t^{-1/2}} \leq 2\operatorname{Tr}(\boldsymbol{H}_T^{1/2})$$

**Proof of Theorem 3**

Since we are preconditioning each layer independently, all three terms in Lemma 1 can be written as summation over the $L$ layers. For simplicity, from now on we omit the summation and the subscript for layers.

For our method, the preconditioner $\boldsymbol{X}_{\boldsymbol{A}_t}$ is as follows,

$$\boldsymbol{X}_{\boldsymbol{A}_t} = \boldsymbol{R}_{\boldsymbol{B}_t}^\dagger(\bar{\boldsymbol{V}}_{\boldsymbol{A}_t} + \rho_{\boldsymbol{A}_t}\boldsymbol{I})^{-1/2}(\boldsymbol{R}_{\boldsymbol{B}_t}^\top)^\dagger$$

We define the unmagnified preconditioner

$$\bar{\boldsymbol{X}}_{\boldsymbol{A}_t} \equiv (\bar{\boldsymbol{V}}_{\boldsymbol{A}_t} + \rho_{\boldsymbol{A}_t}\boldsymbol{I})^{-1/2}$$

Then for the $\boldsymbol{A}$ factor, we have

$$\begin{aligned}
\sum_{t=1}^T \operatorname{vec}(\nabla\boldsymbol{A}_t)^\top\operatorname{vec}(\delta\boldsymbol{A}_t) &= \sum_{t=1}^T \operatorname{Tr}(\nabla\boldsymbol{A}_t^\top\delta\boldsymbol{A}_t) = \eta\sum_{t=1}^T \|\nabla\boldsymbol{A}_t\|_{\boldsymbol{X}_{\boldsymbol{A}_t}}^2 \\
&= \eta\sum_{t=1}^T \|\bar{\nabla}\boldsymbol{A}_t\|_{\bar{\boldsymbol{X}}_{\boldsymbol{A}_t}}^2 \leq 2\eta\operatorname{Tr}((\sum_{i=1}^T \bar{\nabla}\boldsymbol{A}_i^\top\bar{\nabla}\boldsymbol{A}_i)^{1/2}) = O(GT^{1/2}),
\end{aligned} \tag{23}$$

where the inequality comes from Lemma 2 and the fact that

$$\bar{\boldsymbol{X}}_{\boldsymbol{A}_t} = (\bar{\boldsymbol{V}}_{\boldsymbol{A}_t} + \rho_{\boldsymbol{A}_t}\boldsymbol{I})^{-1/2} \preceq (\sum_{i=1}^{t} \bar{\nabla}\boldsymbol{A}_i^\top \bar{\nabla}\boldsymbol{A}_i)^{-1/2}.$$

This completes our proof.

### A.6    PROOF OF THEOREM 4

To prove Theorem 4, we start by bounding the escape mass.

**Lemma 3.** *For the escaped mass $\rho_{\boldsymbol{A}_T}$ in Algorithm 1, we have*

$$\rho_{\boldsymbol{A}_T} \leq \sum_{t=1}^{T} \mathrm{Tr}((\bar{\nabla}\boldsymbol{A}_t)^\top \bar{\nabla}\boldsymbol{A}_t) - \mathrm{Tr}(\bar{\boldsymbol{V}}_{\boldsymbol{A}_T}) = O(G^2 T).$$

**Proof of Lemma 3**

Since $\boldsymbol{P}_{\boldsymbol{A}_t}\boldsymbol{P}_{\boldsymbol{A}_t}^\top \preceq \boldsymbol{I}$, by Ostrowski's inequality we have

$$\lambda_i(\bar{\boldsymbol{V}}_{\boldsymbol{A}_{t-1}}) \geq \lambda_i(\boldsymbol{P}_{\boldsymbol{A}_t}\bar{\boldsymbol{V}}_{\boldsymbol{A}_{t-1}}\boldsymbol{P}_{\boldsymbol{A}_t}^\top)$$

for all $i$. Consequently, we have

$$d_\lambda(\bar{\boldsymbol{V}}_{\boldsymbol{A}_{t-1}}, \boldsymbol{P}_{\boldsymbol{A}_t}\bar{\boldsymbol{V}}_{\boldsymbol{A}_{t-1}}\boldsymbol{P}_{\boldsymbol{A}_t}^\top) \leq \mathrm{Tr}(\bar{\boldsymbol{V}}_{\boldsymbol{A}_{t-1}}) - \mathrm{Tr}(\boldsymbol{P}_{\boldsymbol{A}_t}\bar{\boldsymbol{V}}_{\boldsymbol{A}_{t-1}}\boldsymbol{P}_{\boldsymbol{A}_t}^\top).$$

Thus,

$$\rho_{\boldsymbol{A}_T} \leq \sum_{t=1}^{T} \mathrm{Tr}(\bar{\boldsymbol{V}}_{\boldsymbol{A}_{t-1}}) - \sum_{t=1}^{T} \mathrm{Tr}(\boldsymbol{P}_{\boldsymbol{A}_t}\bar{\boldsymbol{V}}_{\boldsymbol{A}_{t-1}}\boldsymbol{P}_{\boldsymbol{A}_t}^\top)$$

$$= \sum_{t=1}^{T} \mathrm{Tr}((\bar{\nabla}\boldsymbol{A}_t)^\top \bar{\nabla}\boldsymbol{A}_t) - \mathrm{Tr}(\bar{\boldsymbol{V}}_{\boldsymbol{A}_T}) = O(G^2 T).$$

**Proof of Theorem 4**

We prove Theorem 4 by bounding the right-hand side of Lemma 1. Since the first term is a constant and we already bound the second term in Theorem 3, here we only need to bound the third term.

For the third term, we have

$$\|\boldsymbol{A}_t - \boldsymbol{A}_*\|_{\boldsymbol{X}_{\boldsymbol{A}_t}^{-1} - \boldsymbol{X}_{\boldsymbol{A}_{t-1}}^{-1}}^2 = \|(\boldsymbol{A}_t - \boldsymbol{A}_*)\boldsymbol{R}_{\boldsymbol{B}_t}^T\|_{\bar{\boldsymbol{X}}_{\boldsymbol{A}_t}^{-1} - \boldsymbol{Q}_{\boldsymbol{A}_t}\bar{\boldsymbol{X}}_{\boldsymbol{A}_{t-1}}^{-1}\boldsymbol{Q}_{\boldsymbol{A}_t}^T}^2$$

$$\leq D_{\boldsymbol{A}}^2 D_{\boldsymbol{B}}^2 \|\bar{\boldsymbol{X}}_{\boldsymbol{A}_t}^{-1} - \boldsymbol{Q}_{\boldsymbol{A}_t}\bar{\boldsymbol{X}}_{\boldsymbol{A}_{t-1}}^{-1}\boldsymbol{Q}_{\boldsymbol{A}_t}^T\| \leq \mu D_{\boldsymbol{A}}^2 D_{\boldsymbol{B}}^2 \|\bar{\boldsymbol{X}}_{\boldsymbol{A}_t}^{-1} - \boldsymbol{P}_{\boldsymbol{A}_t}\bar{\boldsymbol{X}}_{\boldsymbol{A}_{t-1}}^{-1}\boldsymbol{P}_{\boldsymbol{A}_t}^T\|,$$

where the last inequality comes from our assumption.

Consequently, since

$$\mathrm{Tr}(\bar{\boldsymbol{X}}_{\boldsymbol{A}_t}^{-1}) \geq \mathrm{Tr}(\boldsymbol{P}_{\boldsymbol{A}_t}\bar{\boldsymbol{X}}_{\boldsymbol{A}_{t-1}}^{-1}\boldsymbol{P}_{\boldsymbol{A}_t}^T) \geq \mathrm{Tr}(\bar{\boldsymbol{X}}_{\boldsymbol{A}_{t-1}}^{-1} + c(\rho_{\boldsymbol{A}_{t-1}}^{1/2} - \rho_{\boldsymbol{A}_t}^{1/2})\boldsymbol{I}),$$

we have

$$\sum_{t=1}^{T} \|\bar{\boldsymbol{X}}_{\boldsymbol{A}_t}^{-1} - \boldsymbol{P}_{\boldsymbol{A}_t}\bar{\boldsymbol{X}}_{\boldsymbol{A}_{t-1}}^{-1}\boldsymbol{P}_{\boldsymbol{A}_t}^T\| \leq \sum_{t=1}^{T} \mathrm{Tr}(\bar{\boldsymbol{X}}_{\boldsymbol{A}_t}^{-1} - \boldsymbol{P}_{\boldsymbol{A}_t}\bar{\boldsymbol{X}}_{\boldsymbol{A}_{t-1}}^{-1}\boldsymbol{P}_{\boldsymbol{A}_t}^T) \leq \mathrm{Tr}(\bar{\boldsymbol{X}}_{\boldsymbol{A}_T}^{-1} + c\rho_{\boldsymbol{A}_T}^{1/2}\boldsymbol{I})$$

$$\leq \mathrm{Tr}((\sum_{i=1}^{T} \bar{\nabla}\boldsymbol{A}_i^\top \bar{\nabla}\boldsymbol{A}_i + \rho_{\boldsymbol{A}_T}\boldsymbol{I})^{1/2}) + c\,\mathrm{Tr}(\rho_{\boldsymbol{A}_T}^{1/2}\boldsymbol{I}) \leq (\sqrt{2} + c)\,\mathrm{Tr}((\sum_{i=1}^{T} \bar{\nabla}\boldsymbol{A}_i^\top \bar{\nabla}\boldsymbol{A}_i)^{1/2}),$$

where the last inequality comes from Lemma 3.

Summing up the bound for the second and the third term, we get

$$(2\eta + \frac{\sqrt{2} + c}{\eta}\mu D_{\boldsymbol{A}}^2 D_{\boldsymbol{B}}^2)\,\mathrm{Tr}((\sum_{i=1}^{T} \bar{\nabla}\boldsymbol{A}_i^\top \bar{\nabla}\boldsymbol{A}_i)^{1/2})$$

Choosing $\eta = \mu^{1/2} D_{\boldsymbol{A}} D_{\boldsymbol{B}}$, we have

$$(2 + \sqrt{2} + c)\mu^{1/2} D_{\boldsymbol{A}} D_{\boldsymbol{B}} \operatorname{Tr}((\sum_{i=1}^{T} \bar{\nabla}\boldsymbol{A}_i^{\top} \bar{\nabla}\boldsymbol{A}_i)^{1/2}) = O(D_{\boldsymbol{A}} D_{\boldsymbol{B}} G T^{-1/2}),$$

which completes the proof.

### A.7 TRAINING LOSS CURVE VISUALIZATION

To cross-validate the effectiveness of LoRA-RITE, we plot the training loss curve of each method for the Super-Natural instruction dataset and the OpenBookQA dataset. Figure 2 shows that LoRA-RITE has the lowest training loss, which demonstrates the effectiveness of our method.

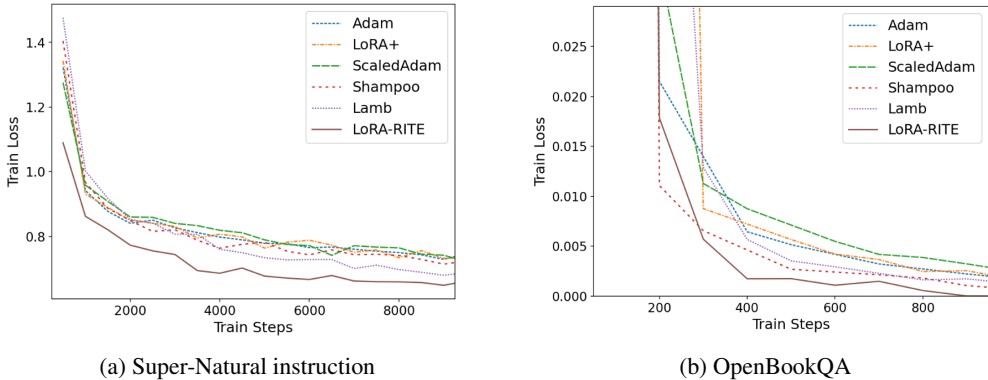

(a) Super-Natural instruction   (b) OpenBookQA

Figure 2: The training loss curve for the Super-Natural instruction dataset and the OpenBookQA dataset.

### A.8 UPDATE MAGNITUDE VISUALIZATION

To visualize the update magnitude of the two LoRA factors, we plot the update norm divided by the weight norm, $\|\delta\boldsymbol{A}\|/\|\boldsymbol{A}\|$ and $\|\delta\boldsymbol{B}\|/\|\boldsymbol{B}\|$.

Figure 3 and Figure 4 show that for conventional optimizers, factor $\boldsymbol{A}$ barely changes, while LoRA-RITE is able to learn the factor $\boldsymbol{A}$ effectively. This demonstrates the importance of transformation invariance.

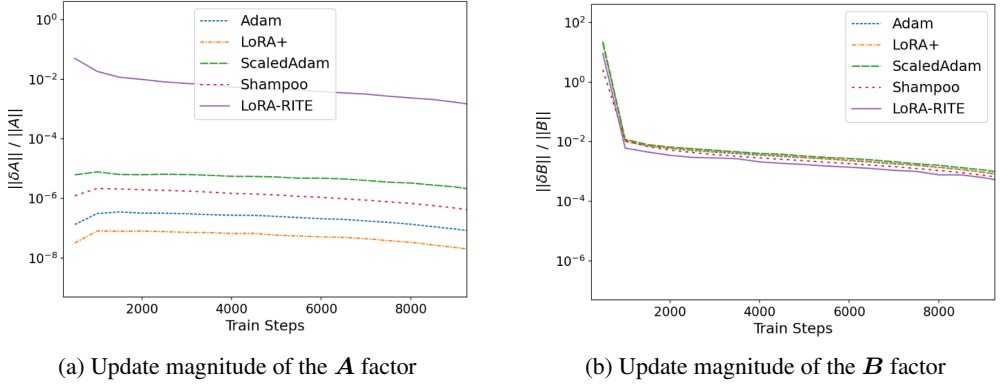

(a) Update magnitude of the $\boldsymbol{A}$ factor   (b) Update magnitude of the $\boldsymbol{B}$ factor

Figure 3: The update magnitude of $\boldsymbol{A}$ and $\boldsymbol{B}$ for the Super-Natural instruction dataset.

### A.9 ABLATION STUDY ON DIFFERENT RANKS

To study the effect of different LoRA ranks, we conduct additional ablation study on different datasets.

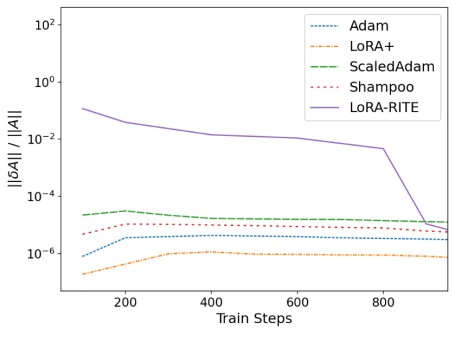

(a) Update magnitude of the $A$ factor

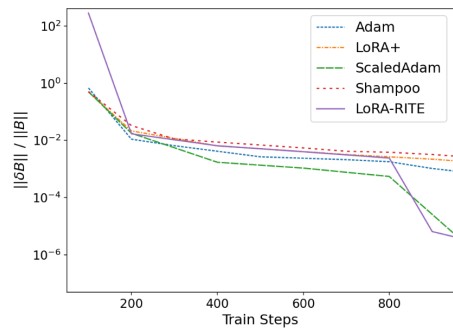

(b) Update magnitude of the $B$ factor

Figure 4: The update magnitude of $A$ and $B$ for the OpenBookQA dataset.

Table 8: Ablation study on different ranks on Gemma-2B on the LLM benchmarking datasets.

| Optimizer | Rank | HellaSwag | ArcChallenge | GSM8K | OpenBookQA | Avg. |
|---|---|---|---|---|---|---|
| Adam | $r = 4$ | 81.83 | 42.32 | 20.92 | 63.0 | 52.02 |
| | $r = 16$ | 83.76 | 45.31 | 24.26 | 64.0 | 54.33 |
| | $r = 64$ | 84.56 | 46.67 | 26.08 | 67.0 | 56.08 |
| ScaledAdam | $r = 4$ | 81.95 | 44.80 | 21.15 | 63.0 | 52.73 |
| | $r = 16$ | 83.52 | 45.22 | 23.96 | 64.8 | 54.38 |
| | $r = 64$ | 84.42 | 48.21 | 26.61 | 67.0 | 56.56 |
| Lamb | $r = 4$ | 86.01 | 46.67 | 25.25 | 67.8 | 56.43 |
| | $r = 16$ | 86.60 | 47.35 | 26.76 | 68.0 | 57.18 |
| | $r = 64$ | 87.83 | 47.53 | 29.04 | 62.8 | 56.80 |
| LoRA-RITE | $r = 4$ | 87.08 | 49.57 | 29.49 | 70.4 | 59.14 |
| | $r = 16$ | 87.28 | 49.06 | 30.10 | 68.8 | 58.81 |
| | $r = 64$ | 87.89 | 49.91 | 31.46 | 68.8 | 59.52 |

As we can see from Table 8, higher rank generally improves LoRA performance, approaching full fine-tuning. This explains why the performance gap between LoRA-RITE and other methods narrows at higher ranks, as they all converge towards the results of full fine-tuning.

Additionally, one can observe that LoRA has inherent regularization properties. As noted in previous research (Chen et al., 2022), this means that sometimes a lower rank can actually lead to better performance. This effect depends on factors like model generalization and training data size. This explains why LoRA-RITE achieves better performance at rank 4 instead of 16 and why Lamb achieves better performance at rank 16 than rank 64.

## A.10   BEST LEARNING RATE FOR DIFFERENT OPTIMIZERS

In Table 9, we list the best learning rate for each optimizer on the LLM benchmarking datasets. We observe that LoRA-RITE and Lamb usually prefer a larger learning rate than the other baselines.

Table 9: Best Learning Rate for Different Optimizers on LLM benchmarking datasets.

| Model | Optimizer | HellaSwag | ArcChallenge | GSM8K | OpenBookQA |
|---|---|---|---|---|---|
| Gemma-2B | Adam | 1e-5 | 5e-5 | 1e-5 | 5e-5 |
| | LoRA+ | 1e-5 | 5e-5 | 1e-5 | 5e-5 |
| | ScaledAdam | 5e-5 | 5e-5 | 1e-5 | 2e-4 |
| | Shampoo | 1e-5 | 5e-5 | 5e-5 | 5e-5 |
| | Lamb | 5e-3 | 5e-3 | 5e-3 | 5e-3 |
| | LoRA-RITE | 2e-4 | 1e-3 | 2e-4 | 2e-4 |

