# OpenReview forum: "LoRA Done RITE: Robust Invariant Transformation Equilibration for LoRA Optimization"
_ICLR.cc/2025/Conference — ICLR 2025 Oral_

### Official Review · Reviewer_vefd · 2024-10-27

**Soundness:** 4
**Presentation:** 4
**Contribution:** 3
**Rating:** 8
**Confidence:** 3

**Summary:**

This paper focuses on improving LoRA, a novel adaptive matrix preconditioning method, and names the proposed method as LoRA-RITE for LoRA optimization. Authors argue that LoRA’s ability to decompose the fine-tuned weight Z in multiple ways is not the best approach for optimization, and they propose a new way for invariant decomposition. Authors have applied the new optimization function, named LoRA-RITE, on Gemma for several datasets and show strong performance on multiple LLM benchmarks.

**Strengths:**

* The overall mathematical proofs are reasonable and appear to be correct.
* With the implementation of LoRA-RITE, authors have shown significant improvement over other optimizers on different language benchmarks.

**Weaknesses:**

I do not have major technical concerns for this paper. It is relatively solid in both performance and implementation analysis. Authors mentioned that they are searching for the learning rate between 2e-6 and 2e-2; it would be interesting to present the best learning rate for different strategies in this work after the search.

**Questions:**

Please check weaknesses section. Authors mentioned that they are searching for the learning rate between 2e-6 and 2e-2; it would be interesting to present the best learning rate for different strategies in this work after the search.

---

> ### Author Response · Authors · 2024-11-25
>
> Thank you for the positive feedback!
>
> > it would be interesting to present the best learning rate for different strategies in this work after the search.
>
> Thank you for the suggestions. In the revised version, we have included the best learning rates for different optimizers on different datasets in Appendix A.9 (Table 9).

---

> ### Author Response · Authors · 2024-12-02
>
> Reviewer vefd,
>
> Thanks again for the positive feedback! We have included the best learning rates for different optimizers on different datasets in the appendix as you suggested.
>
> Additionally, we follow the suggestions of Reviewer zd3u and vefd to make the following modifications.
>
> We include the visualization of the loss curve and the weight updates. This shows that the training loss of LoRA-RITE decreases faster and both the factor A and B can be learned effectively.
>
> We also added experiments to show that that LoRA-RITE performs better than the baselines across different ranks and is competitive with full-finetuning.
>
> Please feel free to let us know if you have any additional questions!
>
> Thanks,
> All authors of Submission 8978

---

### Official Review · Reviewer_wRHA · 2024-11-03

**Soundness:** 4
**Presentation:** 3
**Contribution:** 3
**Rating:** 8
**Confidence:** 4

**Summary:**

This paper points out the issue of standard LoRA training on the decomposed matrices A and B and then enforces the transformation invariant property into optimization, which further improve the representation capability for better performance

**Strengths:**

1. The paper propose a new optimization approach that retains the transformation invariant for the LoRA-type fine-tuning, which is widely used in the large model fine-tuning. Moreover, the paper presented the convergence analysis theoretically.

2. The experimental results demonstrate significant improvement with marginal computation increase, results in a better trade-off compared to all SOTA methods.

**Weaknesses:**

1. As the paper focuses on the optimization, a convergence analysis should be conducted to better justify the proposed method, e.g. the norm of A and B, like in figure 1.

2. The analysis over experimental results are limited, e.g., for some datasets, the proposed method demonstrates significant performance gain compared to LoRA (Adam), what is property of dataset such that the proposed optimization can results such improvement?

**Questions:**

See weaknesses

Questions:

1. As analyzed in at line 322, from time complexity perspective, the proposed method is r times slower than Adam, but in Table 4, it is only 8% slower, it will be interesting to elaborate more the inconsistency.

2. At table 3, when LoRA rank is higher, the improvement over other methods become smaller, I wonder the authors have any insight of that?

3. Why the LAMB paper is cited twich in different way? (line 642, 646)

4. At line 345, LoRA-Rite should be LoRA-RITE for consistency.

---

> ### Author Response · Authors · 2024-11-25
>
> Thank you for the review comments!
>
>
> >  Convergence analysis should be conducted to better justify the proposed method, e.g. the norm of A and B, like in figure 1.
>
> Thank you for the suggestion. We have added visualization of update magnitudes of the proposed method in Appendix A.7 (Figure 3 and 4). Further, to make it more clear to visualize the difference between different methods, we plot the relative update magnitude (update norm divided by parameter norm) which is a better indicator of how much the parameters are updated. The figures show that for conventional optimizers, factor A barely changes, while LoRA-RITE is able to learn factor A effectively.
>
> Additionally, we have added the training loss curves in Appendix A.6. The figures show that the training loss of LoRA-RITE decreases faster than the baseline methods.
>
>
>
> > The analysis over experimental results is limited, e.g., for some datasets, the proposed method demonstrates significant performance gain compared to LoRA (Adam), what is the property of the dataset such that the proposed optimization can result in such improvement?
>
> We are still continually investigating the property of the dataset where LoRA-RITE can have significant performance gain. One observation we have made so far is that LoRA-RITE usually performs better when there is a gap between the baseline methods and full finetuning, while the improvement is usually milder if their performance is already close. This can be seen from the tables below.
>
>
> | Gemma-2B        | CauseEffectClassification | CoreferenceResolution | Title Generation | DatatoText | Global      |
> | --------------- | ------------------------- | --------------------- | ---------------- | ---------- | ----------- |
> | Adam            | 58.93                     | 77.06                 | 51.30            | 55.52      | 50.51/74.54 |
> | ScaledAdam      | 58.71                     | 77.55                 | 51.16            | 55.69      | 49.40/74.01 |
> | Lamb            | 60.97                     | 80.69                 | 52.26            | 55.85      | 53.53/76.43 |
> | LoRA-RITE       | 61.26                     | 82.02                 | 52.26            | 55.98      | 55.11/77.12 |
> | Full Finetuning | 62.77                     | 83.50                 | 52.58            | 56.09      | 60.22/79.19 |
>
> In the table above, we observe a larger gap between full finetuning and lora in CausalEffectClassification, CorefereferenceResolution and Global, and the gap between LoRA-RITE and baselines is also larger on those tasks. Similar observations can be made in the following table, except the second and fourth column where LoRA-RITE outperforms Full Finetuning, which is probably due to some regularization effect.
>
> | Gemma-2B        | Hellaswag | ArcChallenge | gsm8k | openbookqa |
> | --------------- | --------- | ------------ | ----- | ---------- |
> | Adam            | 83.76     | 45.31        | 24.26 | 64.0       |
> | ScaledAdam      | 83.52     | 45.22        | 23.96 | 64.8       |
> | Lamb            | 86.60     | 47.35        | 26.76 | 68.0       |
> | LoRA-RITE       | 87.28     | 49.06        | 30.10 | 68.8       |
> | Full Finetuning | 87.33     | 43.14        | 32.6  | 67.0       |
>
> > As analyzed in at line 322, from time complexity perspective, the proposed method is r times slower than Adam, but in Table 4, it is only 8% slower, it will be interesting to elaborate more on the inconsistency.
>
> The training speed measured in Table 4 includes the time for the model’s forward and backward passes. Since the complexity of the forward and backward passes is at least $\Omega(nm)$, the overhead of LoRA-RITE is relatively small. We have added these explanations to the new version of the paper.

---

> > ### Author Response · Authors · 2024-11-25
> >
> > > In table 3, when the LoRA rank is higher, the improvement over other methods becomes smaller. I wonder if the authors have any insight of that?
> >
> > We have explored the impact of rank extensively, experimenting across various datasets and rank values (see Appendix A.8 for full results). Here's a summary of our findings on all these datasets:
> >
> > 1. Higher rank generally improves LoRA performance, approaching full fine-tuning. This explains why the performance gap between LoRA-RITE and other methods narrows at higher ranks, as they all converge towards the results of full fine-tuning.
> >
> > 2. LoRA has inherent regularization properties. As noted in previous research [1], this means that sometimes a lower rank can actually lead to better performance. This effect depends on factors like model generalization and training data size. This explains why LoRA-RITE achieves better performance at rank 4 instead of 16 and why Lamb achieves better performance at rank 16 than rank 64.
> >
> > [1] Chen, G., Liu, F., Meng, Z., & Liang, S. (2022, December). Revisiting Parameter-Efficient Tuning: Are We Really There Yet?. In Proceedings of the 2022 Conference on Empirical Methods in Natural Language Processing (pp. 2612-2626).
> >
> > > Why is the LAMB paper cited twice in different ways? (line 642, 646)
> > > At line 345, LoRA-Rite should be LoRA-RITE for consistency.
> >
> > Thanks! We have fixed the typos in the revision.

---

> ### Author Response · Authors · 2024-12-02
>
> Dear Reviewer wRHA,
>
> Thanks again for your constructive suggestions!
>
> We follow your suggestion to include the visualization of the loss curve and the weight updates.
> We also added experiments to show that that LoRA-RITE performs better than the baselines across different ranks and is competitive with full-finetuning.
>
> We believe the addition of these results further verifies the effectiveness of LoRA-RITE and will be very helpful to the readers.
>
> Please feel free to let us know if you have any additional questions!
>
> Thanks,
> All authors of Submission 8978

---

### Official Review · Reviewer_zd3u · 2024-11-08

**Soundness:** 4
**Presentation:** 3
**Contribution:** 4
**Rating:** 10
**Confidence:** 3

**Summary:**

The paper introduces an enhancement to the existing LoRA method. LoRA, which stands for Low-Rank Adaptation, is a parameter-efficient fine-tuning technique widely employed in training Large Language Models. However, a significant limitation of LoRA is that matrix $A$ is typically much larger than matrix $B$, leading to reduced training efficiency of LoRA. The authors leverage the concept of transformation invariance to formally capture this phenomenon, demonstrating that LoRA weight updates remain consistent across different decomposition results. They further define a weaker version, termed "scalar scale invariance," which only examines whether decompositions with varying magnitude distributions in matrices $A$ and $B$ yield different weight updates. The authors demonstrate that transformation invariance is crucial for efficient feature learning, propose a method to preserve this invariance while considering optimizer implementation, and conduct experiments across various models to illustrate the advantages of their approach.

**Strengths:**

The authors take a significant step beyond simple intuitions by delving into the core of training efficiency, proposing a novel approach to fully achieve transformation invariance in LoRA models. For soundness, they provide rigorous proofs for all mathematical statements in the paper and conduct extensive experiments across various LoRA advancement methods and benchmarks to demonstrate the superiority of their method. In terms of contribution, given that LoRA is a widely-used training method and their work is computationally inexpensive, individuals training Large Language Models and potentially other AI models can greatly benefit from their advancements.

**Weaknesses:**

The paper lacks visual illustrations of loss curves to cross-validate the effectiveness of their method in accelerating convergence. Although they state that matrix $A$ remains nearly identical during training, they do not provide visual evidence of how the magnitude of $A$ updates after applying their method, which could further validate its effectiveness. Additionally, the authors do not address potential numerical instability issues. Specifically, their algorithm involves inverting the matrix $R_A$, which could be zero at the initial training step. The conclusion and limitations section is somewhat lacking, as it only discusses the limitations of the base LoRA method rather than potential limitations of their proposed method.

**Questions:**

Will your method be susceptible to numerical instability? If not, why? If so, how will you address this issue?

In terms of contributing to the LLM community, open-sourcing the related code would be highly appreciated.

---

> ### Author Response · Authors · 2024-11-25
>
> Thank you for the detailed review!
>
> > The paper lacks visual illustrations of loss curves to cross-validate the effectiveness of their method in accelerating convergence.
>
> Thank you very much for the suggestion. We have added the training loss curves in Appendix A.6 (Figure 2). The figures show that the training loss of LoRA-RITE decreases faster than other methods.
>
> > The authors do not provide visual evidence of how the magnitude of $A$ updates after applying their method
>
> Thanks for the suggestion! We have added visualizations of update magnitudes of Lora-RITE to Appendix A.7 (Figure 3 and 4). Further, to make it more clear to visualize the difference between different methods, we plot the relative update magnitude (update norm divided by parameter norm) which is a better indicator of how much the parameters are updated. The figures show that for conventional optimizers, factor A barely changes, while LoRA-RITE is able to learn factor A effectively.
>
> > The authors do not address potential numerical instability issues. Specifically, their algorithm involves inverting the matrix $R_A$, which could be zero at the initial training step.
>
> We thank the reviewer for the question. We would like to clarify that at the initial training step it is $R_B$ that is 0 instead of $R_A$.
>
> There are two places in the algorithm that could have potential numerical instability and we did carefully address the numerical issues in our implementation as discussed below:
>
> 1. For inverting the matrix $R_A$ and $R_B$, instead of directly inverting $R_A$, we apply the pseudoinverse (we have added this note to the revised paper). At the initial training step,  $R_B$ is 0, and so is the gradient of $A$. After multiplying with the pseudoinverse of $R_B$, we get that the unmagnified gradient of $A$ is also 0 to avoid NaNs.
>
> 2. For computing the inverse square root of $\bar{V_A}$  and $\bar{V_B}$, we add a small $\epsilon I$ to $\bar{V_A}$  and $\bar{V_B}$ respectively, before taking the inverse square root. Since  $\bar{V_A}$  and $\bar{V_B}$ are already unmagnified, this does not impact transformation invariance.
>
> We have included these details in the revision.
>
> > The conclusion and limitations section is somewhat lacking, as it only discusses the limitations of the base LoRA method rather than potential limitations of their proposed method.
>
> Thanks for the suggestion. We have added the potential limitations of LoRA-RITE to the revised version: The work focuses on addressing transformation invariance when the optimization problem can be written in the form of f(AB), and this assumption may not hold for parameter-efficient structures beyond LoRA.
>
> > In terms of contributing to the LLM community, open-sourcing the related code would be highly appreciated.
>
> Thank you for pointing out this. We plan to release the related code after it passes internal review.

---

> > ### Comment · Reviewer_zd3u · 2024-12-03
> >
> > Thank you for your comprehensive explanation! The supplementary visualizations are sufficiently detailed to substantiate the validity of your methodology, and your approach to mitigating numerical instability makes sense to me. I am now convinced that your method has the potential to make a significant contribution to the LLM community, particularly when your code is made open-source. Given that all previously identified weaknesses have been satisfactorily addressed, I am pleased to increase my score to 10.

---

> ### Author Response · Authors · 2024-12-02
>
> Dear Reviewer zd3u,
>
> Thanks again for your insightful suggestions.
>
> We follow your suggestion to include the visualization of the weight updates.
> Additionally, the extra experiments asked by Reviewer wRHA shows that LoRA-RITE performs better than the baselines across different ranks and is competitive with full-finetuning.
> We believe these results further verify the effectiveness of LoRA-RITE.
>
> We also improve the writing to include details on addressing potential numerical instability issues and to discuss the limitation of our method. We believe these modifications will greatly help the readers.
>
> Please feel free to let us know if you have any additional questions!
>
> Thanks,
> All authors of Submission 8978

---

### Meta-Review · Area_Chair_vwZ3 · 2024-12-23

**Metareview:**

This study proposes an adaptive matrix preconditioning method for LoRA optimization to achieve transformation invariance, which can mitigate the dependence on how LoRA factors are scaled and rotated to avoid sub-optimal solutions and improve the representation capability. All reviewers recognize the quality of this study. The concerns about visual illustration and convergence analysis are addressed by the authors. The AC agrees with the reviewers and recommends Accept.

**Additional Comments On Reviewer Discussion:**

The major concerns of reviewers are visual illustration and convergence analysis. They are addressed by the authors by providing training loss curves and update magnitudes during the rebuttal period.

---

### Decision · Program_Chairs · 2025-01-22

Accept (Oral)